# Direct MRI of collagen

Jason Daniel Van Schoor, Markus Weiger*, Emily Louise Baadsvik,
Klaas P Pruessmann

Institute for Biomedical Engineering, ETH Zurich and University of Zurich, Zurich,
Switzerland

## eLife Assessment

This **fundamental** work substantially advances our understanding of a major research question: whether collagen can be directly imaged with MRI. The evidence supporting the conclusion is **compelling**, with methods, data, and analyses that are more rigorous than those currently considered state-of-the-art. The work will be of high interest to MR physicists and clinicians, as collagen is the most abundant protein in the human body and plays an essential role in health.

**Abstract** Collagen is the most abundant protein in the human body and has an important role in healthy tissue as well as in a range of prevalent diseases. Medical research and diagnostics, hence, call for means of mapping collagen in vivo. Magnetic resonance imaging (MRI) is a natural candidate for this task, offering full 3D capability and versatile contrast non-invasively. However, collagen has so far been invisible to MRI due to extremely short lifetime of its resonances. Here, we report the direct imaging of collagen in vivo by magnetic resonance on the microsecond scale. The dynamics of resonance signals from collagen were first assessed in samples of bovine tendon and cortical bone. On this basis, imaging was performed at echo times down to 10 microseconds, yielding collagen-specific depiction by echo subtraction. The same approach was then extended for use in vivo, enabling direct collagen imaging of a human forearm. This capability suggests significant promise for biomedical science and clinical use.

**\*For correspondence:**
weiger@biomed.ee.ethz.ch

**Competing interest:** The authors declare that no competing interests exist.

## Introduction

Accounting for about 30% of protein mass, collagen is ubiquitous in vertebrates and plays an integral role in providing tissue strength, flexibility, and support (*Sandhu et al., 2012*). It is a key constituent of the extracellular matrix in structures, such as bone, skin, blood vessels, cartilage, ligaments, tendon, and dentine. Variations in collagen content or structure have a major role in aging and are associated with pervasive diseases, such as arthritis and fibrosis. Osteo- and rheumatoid arthritis, in particular, are musculoskeletal conditions that affect a large part of the global population (*Cieza et al., 2020*). Rheumatoid arthritis is an autoimmune disorder in which the immune system targets the joints, causing inflammation and a thickening of the tissues lining the joints (*Mohammed et al., 2020*). Osteoarthritis is a degenerative joint disease involving the breakdown of cartilage. Both result in collagen degradation and lead to pain, swelling, and reduced joint mobility (*Mohammed et al., 2020*; *Ouyang et al., 2023*). In contrast, fibrosis is the excessive and uncoordinated deposition of collagen and other matrix proteins and is linked to an impaired healing response to infection or injury (*Coelho and McCulloch, 2016*). It affects most organs, can lead to chronic organ failure, and contributes significantly to global mortality (*Mutsaers et al., 2023*). In aging, finally, modifications to the biomechanical and biophysical characteristics of tissue result from the consolidation of collagen (*Wilson et al., 2014*). Consequently, collagen loses its pliability, malleability, and susceptibility to digestion, thereby compromising the functionality of related organs and tissues.

**eLife digest** Magnetic resonance imaging (MRI) is a well-established clinical technique for examining the body non-invasively. It uses magnetic fields and radio waves to disturb hydrogen atoms in the body, which then emit signals as they return to a normal state. A computer analyses these signals to create detailed images of different tissues inside the body.

Because these signals have a limited lifetime, the duration of the imaging process is critical in determining which tissues can be detected. Conventional MRI primarily measures signals from bulk water in soft tissues, which decay over tens to hundreds of milliseconds. However, a substantial fraction of signals in the body decays on the microsecond timescale, making them inaccessible to standard MRI methods.

One important source of such rapidly decaying signals is collagen, the most abundant protein in the human body and a key structural component of tissues including skin, cartilage, tendons and bone. Because of its extremely short signal lifetime, collagen has traditionally been assessed only indirectly through MRI of the surrounding water.

Direct collagen MRI could offer greater specificity than indirect approaches and support both research and clinical applications. For example, it could improve understanding of tissue changes associated with disease and injury or enable bone density measurements without exposure to ionising radiation.

To find out if MRI could image collagen directly, van Schoor et al. used a combination of recently developed custom hardware and specialised imaging methodology designed to detect and spatially encode the extremely short-lived collagen signal. The approach was first validated in bovine tendon and bone samples and subsequently extended to imaging of the human forearm.

The images obtained not only visualised collagen directly but also captured the rapid decay of its signal over time. This demonstrates that direct MRI of collagen is indeed feasible.

Direct MRI of collagen could have important applications in fields such as musculoskeletal medicine, tissue engineering, and fibrosis research, where collagen content and organisation are central to tissue function and pathology. Although the current method still relies on custom-built hardware, these findings provide a foundation for developing clinical MRI systems capable of imaging rapidly decaying signals in a wider range of tissues, diseases and patient populations. With further refinement, this approach could complement existing imaging techniques, provide new non-invasive insights into tissue structure, and potentially enable direct MRI of other macromolecules.

The critical importance of collagen prompts the need for advanced imaging tools to aid in diagnostics and research of collagen-related diseases. Methods such as X-ray diffraction (*Brodsky et al., 1988*) and electron microscopy (*Hulmes et al., 1981*) are used to evaluate collagen structure and organization on the molecular scale, whilst mass spectrometry can be used to quantify and distinguish different types of collagen (*van Huizen et al., 2020*). Ex-vivo visualization of bulk collagen is typically achieved using optical techniques like fluorescence and visible light microscopy (*Bielajew et al., 2020*). Other optical modalities, such as second-harmonic-generation microscopy, have been used to evaluate fibrillar collagen in human tissue in vivo, but only for superficial structures due to limited penetration depth (*Mostaço-Guidolin et al., 2017*). None of these modalities appears to be suited for non-invasive routine use in vivo.

Towards in vivo collagen mapping, MRI is an attractive candidate because it achieves 3D coverage and variable contrast non-invasively and without the use of ionizing radiation. Capturing collagen by [1]H MRI requires recording of nuclear magnetic resonance (NMR) signals from the hydrogen atoms of the collagen molecule. Due to the macromolecular nature of collagen, the [1]H nuclei experience strong dipolar coupling and related line broadening (*Mroue et al., 2015*; *Eliav and Navon, 2002*; *Ong et al., 2012*). In NMR studies, linewidths can be narrowed by radiofrequency (RF) decoupling or magic-angle spinning (*Laws et al., 2002*). These approaches reveal distinct resonances associated with specific binding sites in the collagen molecule, providing insight into its molecular structure. However, such methods are not suitable for in vivo imaging. As a consequence, collagen signal on MRI systems exhibits a single, very rapid, decay with a relaxation time constant, $T_2$, of ~10–20 μs ($T_2^*$ in collagen macromolecules is dominated by $T_2$, whereas $T_2'$ associated with inhomogeneous broadening

is negligible; therefore, within the scope of this work, $T_2$ is always used for the sake of simplicity) (*Seifert et al., 2014*; *Seifert and Wehrli, 2016*; *Horch et al., 2010*; *Fantazzini et al., 2003*; *Edzes and Samulski, 1978*). Well below the timescales of current MRI techniques, signal lifetimes this short have made collagen effectively MR-invisible (*Seifert et al., 2014*; *Ma et al., 2016*; *Guo et al., 2024*). Rather, indirect MRI approaches have been employed to study collagen, including the use of collagen-targeted contrast agents (*Caravan et al., 2007*), magnetization transfer (MT) imaging (*Ma et al., 2020*), and collagen-bound water imaging through dedicated short-$T_2$ techniques with ultra-short or zero echo time (TE) (*Seifert et al., 2014*; *Chen et al., 2015*; *Surowiec et al., 2021*; *Weiger et al., 2013*).

In this work, we report direct collagen MRI, matching rapid signal decay by encoding and detection on the scale of tens of microseconds. Free induction decays (FIDs) are recorded from collagen-rich tendon and cortical bone samples, featuring strong, rapidly decaying collagen signals that are readily distinguished from longer-lived bound- and free-water components. Spatial encoding of data taken with different timings, followed by subtraction, yields collagen images of tissue samples and a human forearm in vivo.

## Results
### Collagen magnetic resonance signal detection and behavior

To explore the collagen MR signal, FIDs were acquired from collagen-rich samples of bovine tendon and bone before and after treatment for removing the sources of water signal (*Ma et al., 2016*)

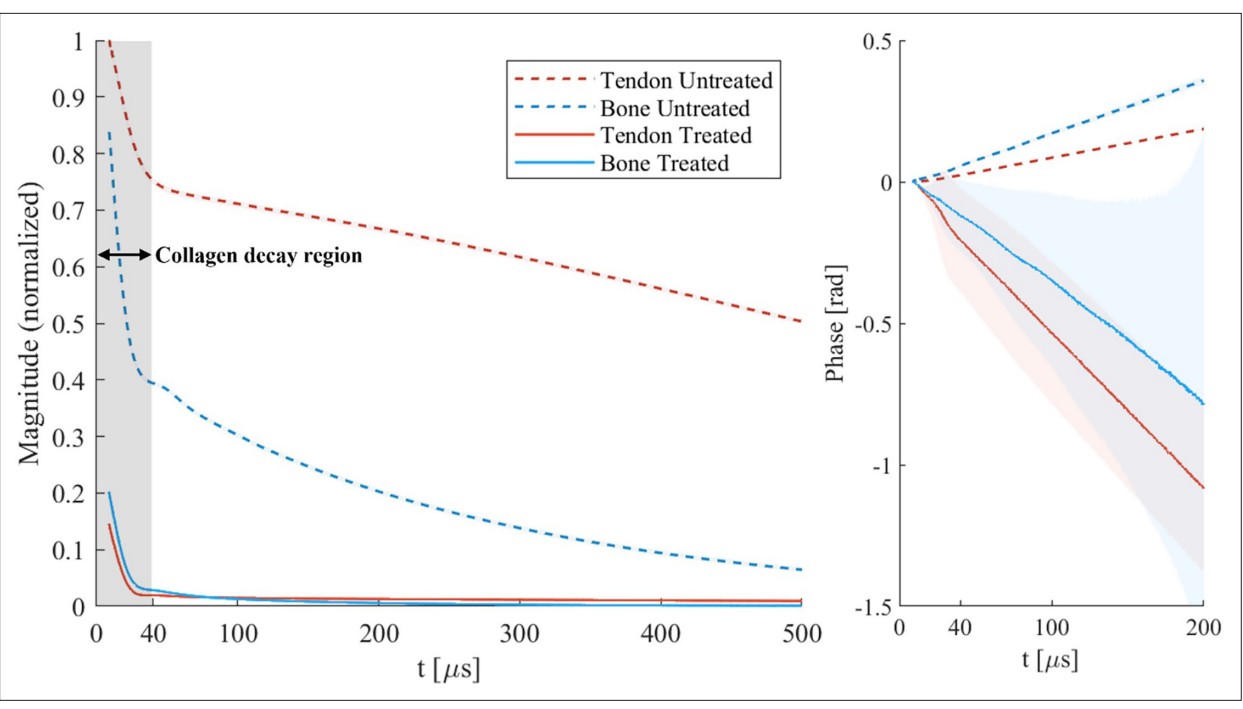

**Figure 1.** Free induction decay (FID) signals of tendon and cortical bone samples before and after treatment with a procedure to remove the water signal. The magnitude has been normalized by the maximum signal observed in the unprepared tendon. The shaded areas around the lines indicate the 90% central ranges of the averaged data, which are generally very narrow and only large for the phases of the treated samples. A rapidly decaying signal, attributed chiefly to collagen, is observed on top of the slower-decaying collagen-bound water signal for untreated samples. Much longer decaying components from free water and fat are also present but hardly differentiable at this timescale. The collagen component appears to have completely decayed after ~40 μs. The treated bone signal decays into the noise floor. The treated tendon signal remains above the noise floor due to the presence of longer-living fat signal in the tissue. The distinct bump observed in the magnitude data (for treated and untreated bone, and treated tendon) is attributed to an oscillation stemming from dipolar coupling of the collagen protons. The reduced amplitude of the rapidly decaying component after treatment is believed to be a result of H-D exchange of exchangeable protons on collagen and other molecules with short-lived signals. The variation in phase accrual observed before and after treatment indicates chemical shift differences of the signal contributors that persist post-treatment. The marked collagen decay region indicates the maximum time range available for spatial encoding in direct collagen imaging. In contrast, imaging based on collagen-bound water can utilize the full plotted range.

(including H-D exchange and freeze drying, see Methods). As shown in *Figure 1*, the signals exhibit a rapidly decaying component, which has effectively vanished 40 µs after RF excitation. This signal component is chiefly attributed to collagen based on its agreement with previously reported signal behavior (*Ni et al., 2007*; *Nyman et al., 2008*; *Schreiner et al., 1991*; *Funduk et al., 1984*), the correspondence of the estimated collagen signal fraction to the observed signal contributions (see Appendix 1), and the fact that magnetization exchange between collagen and bound water operates in the slow regime permitting their independent observation (see Discussion). This last point is supported by the observation that there is hardly any change in decay rates of the shortest-lived component between treated and untreated samples. In the untreated samples, the rapidly vanishing signal is accompanied by a much slower decaying component that is no longer observed after treatment. This component is attributed to collagen-bound water. The modified signal composition is also observed in the accrued signal phase, the slope of which changes after treatment, indicating dominance by another component of different chemical shift. The different signal behavior of the collagen-bound water in bone and tendon is evidence of different tissue structure. In the treated bone, the signal decays into the noise floor. In the treated tendon, a low-amplitude, apparently constant signal is observed, which remains above the noise floor for the displayed duration and which we attribute to fat. A distinct bump is observed in the magnitude of the FIDs at ~20–40 µs for all samples except the untreated tendon. It reflects a subtle oscillation stemming from dipolar coupling of the collagen protons with an interaction in the kHz range (*Mroue et al., 2015*). Note that the applied treatment also reduces the amplitude of the rapidly decaying component, indicating that the H-D exchange part of the treatment has also occurred in components with short-lived MR signals due to the presence of exchangeable protons on macromolecules (*Englander et al., 1996*).

To more quantitatively investigate the features of the observed FIDs, a representative signal model was found permitting a fit of the decay curves, including the observed bump, with a limited number of components (see Appendix 2). The fitted components describe the overall signal behavior, but do not unambiguously reflect distinct proton pools due to restricted model complexity.

Overall, the observed signal contributions from collagen and water are of comparable order, indicating that the available level of collagen signal can be expected to provide a suitable basis for direct collagen imaging at useful signal-to-noise ratio (SNR). That said, the rapid signal decay poses a significant challenge compared with use of the longer-lived signal of collagen-bound water.

## Direct collagen MR images are produced using two ultra-short echo times

To enable MRI based on the extremely short-lived collagen signal, very short TEs and rapid spatial encoding were achieved by means of advanced, custom short-$T_2$ technology, including high-performance RF and gradient hardware as well as dedicated imaging methods. Short-$T_2$ imaging was performed at multiple, increasing TEs in the untreated and treated collagen-rich samples, thus enabling observation of the signal behavior also with spatial localization.

*Figure 2* demonstrates direct collagen imaging, with resulting images displayed for selected early TEs. For the two shortest TEs, all samples are depicted at substantial signal intensity. A decrease in intensity is observed for increasing TEs in all samples, in agreement with the early decay of the collagen signal observed in the FIDs. However, the intensity decrease is less obvious in the untreated samples because the water signal dominates. In both treated samples, the image intensity is significant at the earliest TE of 10.4 µs and rapidly decreases to be nearly unobservable by a TE of 35.4 µs. In addition, obvious blurring effects are visible as compared to the untreated samples, indicating dominance of signals with very short $T_2$s for which the rapid decay limits the spatial resolution. These findings are consistent with the effective resolution associated with $T_2$ blurring calculated using the method by *Froidevaux et al., 2020* (Appendix 3). Overall, the observations support the interpretation that the images of the treated samples primarily show collagen.

In *Figure 3*, the decay of collagen signal is observed with spatial localization. ROIs are drawn over the same region for all acquired TEs, and the mean signal intensity is plotted for each sample as a function of TE. Overall, the observed signal characteristics match those of the magnitude of the FIDs in *Figure 1*. In particular, the rapid initial signal decay and dipolar oscillation are present.

To further support image interpretation and to investigate the potential of in vivo collagen-specific MRI, isolation of the collagen signal was targeted by subtraction of images with different

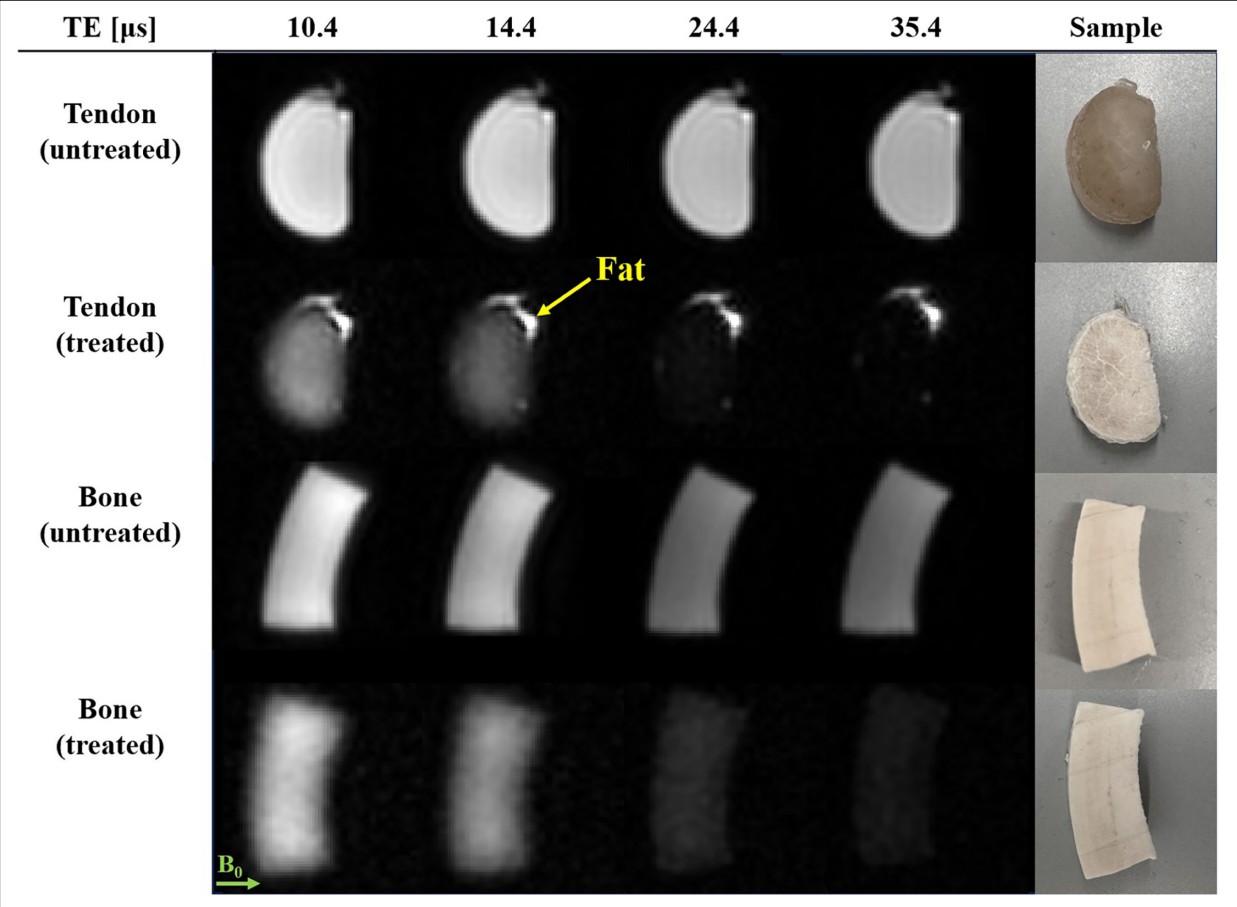

**Figure 2.** Direct magnetic resonance imaging (MRI) of collagen. From an image series of collagen-rich tissue samples with increasing echo times (TEs), four examples at early TEs are shown. The magnitudes are normalized by the maximum signal in the shortest-TE image of the respective sample. A decrease in image intensity is observed with increasing TE, reflecting the decay of collagen signal. In the treated samples, the signal has virtually completely decayed by TE = 35.4 µs. The effect is less obvious in the untreated samples due to strong background signal from water contained in the samples. In the treated tendon sample, the bright fat signal appears constant due to negligible signal decay over the given timescale.

TEs (*Li et al., 2012*; *Johnson et al., 2017*; *Szeverenyi and Carl, 2012*; *Rahmer et al., 2006*; *Ma et al., 2021*). This approach is justified by the observation that the signals of the investigated tissue samples exhibit decay times for collagen that are clearly distinct from those of the longer-lived contributions. The shortest-TE image was used to provide the highest contribution from collagen. An image at slightly longer TE was selected to exhibit significant decay of the collagen signal but negligible change of the longer-lived signal components. In this way, high specificity for collagen is achieved at useful sensitivity. *Figure 4* displays the results of selective collagen imaging for untreated and treated samples. In the difference images, the longer-lived water and fat signals are suppressed, leaving only the short-lived collagen signal component. For the treated samples, the subtraction has only a small effect, confirming that the treatment largely removes the longer-living signal contributions and the images mainly show collagen. In tendon, the subtraction removes the fat signal, which was not affected by the treatment. In the untreated samples, subtraction successfully removes the long-lived signals, indicated by the blurrier appearance of the difference images. The slightly sharper depiction of the untreated versus treated samples can be explained by a small amount of residual longer-lived signal. These findings are supported by simulations of the full imaging and subtraction procedure based on the initially observed signal characteristics (see Appendix 4). Notably, in the treated tendon, a bright band is observed in the difference image suggesting a change in local tissue structure. A deeper analysis of this observation can be found in Appendix 5.

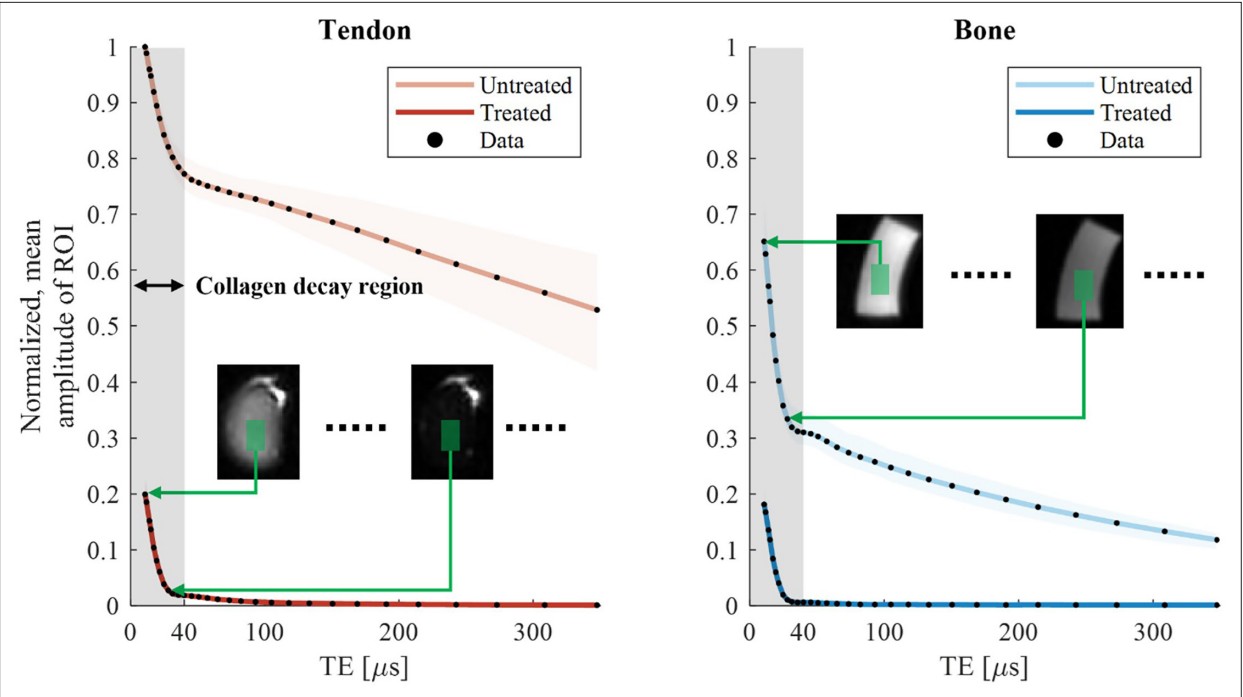

**Figure 3.** Decay of collagen signal observed with spatial localization by imaging at multiple echo times (TEs). The magnitude of the mean signal in a region of interest (ROI) in the bone and tendon samples is plotted as a function of TE. The rapidly decaying collagen signal is observed in both the treated and untreated samples. These plots are equivalent to the free induction decay (FID) signal magnitudes shown in *Figure 1*, including the dipolar oscillation in bone and treated tendon at the interval of ~20–40 µs. However, with imaging, the signal can be observed at specific locations. The shaded areas around the lines indicate the 90% central ranges of the averaged data points.

## Direct collagen MRI can be performed in vivo

For in vivo demonstration, the experimental protocol underlying *Figure 4* was modified for imaging of a human forearm. *Figure 5* shows the two resultant short-TE images and their subtraction indicating collagen content distribution. In the image with slightly longer TE, reduced intensity can be readily observed in cortical bone. Consequently, the difference image shows positive contrast for bones. Furthermore, tendon and skin are visible. In contrast, tissues with long-lived signals, such as bulk water in muscle, bone marrow, and other fatty structures, are largely suppressed in the difference image. Muscle exhibits a higher intensity after subtraction than signal-void locations (outside the arm) and bone marrow, suggesting that rapidly decaying signal contributions are also detected in muscle tissue. These contributions may arise from condensation of dense connective tissue at the muscle margins, forming tendons or aponeuroses or even from non-collagenous macromolecules, such as other proteins and phospholipids (reported $T_2$ of ~16–25 µs) (*Edzes and Samulski, 1978*; *Sobol et al., 1986*; *Belton et al., 1972*; *Ribeiro et al., 1984*; *Radoicic et al., 2014*).

## Discussion

The results obtained in this study confirm that direct collagen imaging is possible. In FIDs, collagen signal was identified as a very rapidly decaying component. Spatial resolution of this decay was achieved using advanced, custom short-$T_2$ technology and a multi-TE imaging strategy. Collagen signals were captured by the earliest TEs in the series, and image subtraction yielded a collagen-selective depiction both in tissue samples and in vivo.

The FIDs shown in *Figure 1* reproduce the signal behavior of similar tissues in previous works (*Ni et al., 2007*; *Nyman et al., 2008*; *Schreiner et al., 1991*; *Funduk et al., 1984*) where contributions with distinct $T_2$s are assigned to macromolecular protons and bound-water protons, respectively. The independent observation of macromolecular and bound-water signals is dependent on the rate of magnetization transfer between the two pools. Using the Bloch-McConnell model (*McConnell, 1958*) outlined by *Vallurupalli, 2009*, the magnetization exchange regime is slow if $k << |\Delta R|$, where $k$ is

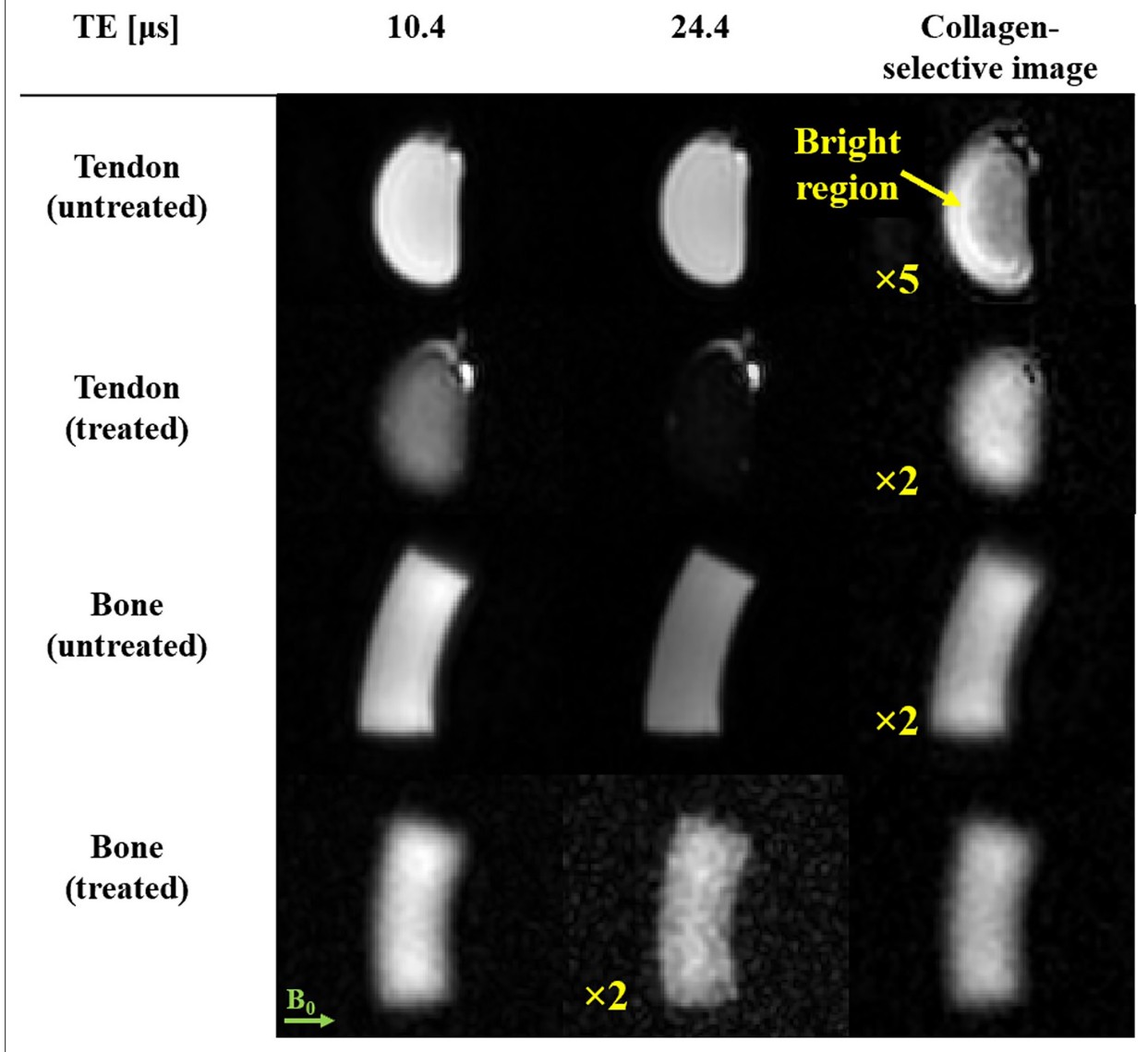

**Figure 4.** Selective imaging of collagen in tendon and bone samples. Isolation of the collagen signal is achieved by subtraction of the shortest-echo time (TE) image (10.4 µs) and an image with slightly longer TE (24.4 µs). The image intensities have been normalized by the maximum signal observed in the shortest TE image of the respective sample. The bright fat signal in the tendon has been clipped to visualize the collagen in the non-difference images. The images have been scaled, as indicated, for visualization purposes. In the difference images, long-lived water and fat signals are suppressed, leaving only the short-lived components, which are attributed chiefly to collagen. For the treated samples, the effect of subtraction is negligible. The fat signal in treated tendon persists and is removed after subtraction, indicating that the treatment does not impact the fat and that longer-lived signals are suppressed by the subtraction procedure. The difference images of the untreated samples exhibit less $T_2$ blurring than the treated samples due to residual signal contamination from the collagen-bound water signal (Appendix 4). In the untreated tendon, the subtraction yields a bright region that is not visible in the treated counterpart. This suggests an effect of the treatment on the samples that is observable at the collagen timescale.

the magnetization exchange rate and $\Delta R$ is the difference in the relaxation rates between the two proton pools. With reported values of $k$ for tendon and cortical bone on the order of 1–80 s⁻¹ (*Eliav and Navon, 2002*; *Ma et al., 2018*) and estimated $\Delta R$ on the order of $10^4$–$10^5$ s⁻¹, the magnetization exchange regime is slow and thus macromolecular and bound-water signals are indeed separably observable.

We assign the rapidly decaying macromolecular component primarily to collagen according to the expected signal fraction estimated in Appendix 1. This component has a $T_2$ on the order of 10 µs in agreement with previous NMR work (*Seifert et al., 2014*; *Seifert and Wehrli, 2016*; *Horch et al., 2010*; *Fantazzini et al., 2003*; *Edzes and Samulski, 1978*). Additionally, small signal contributions

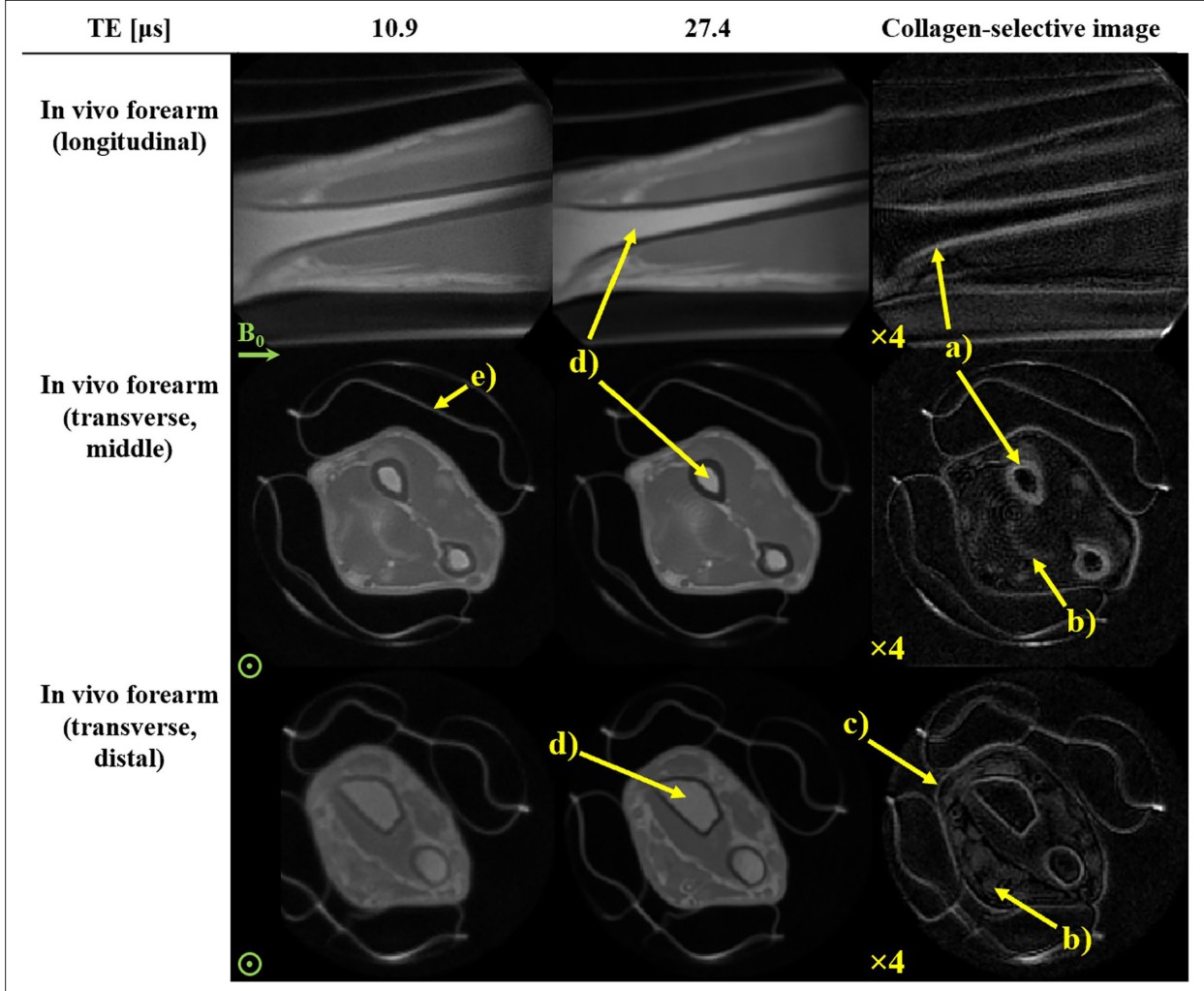

**Figure 5.** Direct collagen magnetic resonance imaging (MRI) of a right human forearm in vivo. Three views of the forearm are shown at two short-echo time (TE) acquisitions along with the collagen-selective difference image. Images have been normalized according to the maximum signal observed in the earliest TE image of the displayed slices and scaled as indicated. Note that the difference in contrast of the two raw transverse views is due to variations in transmit sensitivity of the radiofrequency (RF) coil. The difference images show collagen-rich anatomy, such as (**a**) cortical bone, (**b**) tendon, (**c**) skin and subcutaneous tissue, whereas longer-lived signals, such as from (**d**) trabecular bone marrow are suppressed. Signal from padding, (**e**), is also captured. A slight ringing artifact (typical of high-bandwidth radial acquisitions) is observed in the transverse view.

from other macromolecules with $T_2$s of this order are also expected to be present (*Seifert and Wehrli, 2016*). The difference in magnitude after treatment of the shortest-$T_2$ component may be a result of H-D exchange with protons in hydroxyl groups or amines of collagen, or on other macromolecules, such as hydroxyapatite in bone (*Englander et al., 1996*). The rate of decay of the collagen signal between treated and untreated samples appears largely unchanged, further supporting the claim that magnetization transfer between collagen and collagen-bound water has little effect on the observability of the collagen signal. The distinctive bump occurring at ~30 µs is believed to be a dipolar oscillation from interactions between collagen protons. Multiple dipolar coupling phenomena exist within collagen (*Brown and Spiess, 2001*), and a superposition of these interactions is observed here. Isolating specific interactions is hampered by low spectral resolution, which is itself a consequence of dipolar coupling. Furthermore, magnetization transfer between collagen protons occurs in the fast regime, and impacts the observed signal decay (*Eliav and Navon, 2002*). The slower decaying signals, with $T_2$s on the order of 100s of microseconds, in the untreated samples are attributed to collagen-bound water. The bound-water signal has completely vanished in the treated samples and only the collagen component remains, suggesting that the treatment to remove water signal was successful. The weak, apparently constant signal component that remains above the noise floor in the

treated tendon is assumed to be from residual fat, which is unaffected by the treatment (*Englander et al., 1996*) and decays much slower than would be observable at this timescale. Using a model with a relatively small number of components, the FID signals could be well fitted, including the dominating collagen component with dipolar oscillation (see Appendix 2). However, the terms describing the dipolar oscillation are rather a simplification of the underlying interactions and on this basis, and extracted signal components from the model are not assigned to specific proton pools. Despite this, excluding the dipolar oscillation and describing the signal as a sum of basic decay functions could still be a viable alternative to extract tissue-specific amplitude components, as suggested previously (*Ni et al., 2007*; *Nyman et al., 2008*; *Schreiner et al., 1991*; *Funduk et al., 1984*). Such efforts have been reported for cortical bone (*Horch et al., 2010*) and myelin (*Baadsvik et al., 2024*) using spectral- and time-domain modeling, respectively.

The rapid collagen signal decay during an imaging readout is prone to causing significant image blurring (*Weiger and Pruessmann, 2019*) (see Appendix 3). For the current imaging protocols, exponential decay with $T_2$=12 μs for collagen leads to an effective resolution of 2.2 mm as compared to 1.6 mm without decay (*Froidevaux et al., 2020*). To contain the blurring effect, short observation times must be achieved by rapid spatial encoding using strong gradients, which comes at an intrinsic SNR expense compared with acquisition at lower bandwidth. The effective resolution can be further improved to some degree by increasing nominal resolution or using longer dead times, albeit at further expense in SNR efficiency (*Froidevaux et al., 2020*). Hence, a reasonable compromise between resolution, SNR, and scan time must be found, particularly for in vivo applications. Regarding the critical role of SNR, the results of this work suggest that the large collagen content of tissues of interest affords considerable SNR permitting detailed depiction.

Rapid RF switching and spatial encoding with custom hardware has enabled a multi-TE experiment with very short, continuously selectable TEs. The absence of collagen signal after ~35 μs (*Figure 2*) suggests that using TE longer than this is not beneficial for studying collagen directly. In practice, fewer TEs could be used in the multi-TE acquisition to reduce scan time. Alternatively, additional TEs could be acquired at earlier intervals to improve the temporal resolution of signals decaying within these intervals. This could provide the insight required to isolate signal components with similar decay rates to collagen, such as hydroxyapatite in bone.

Both the multi-TE plots and the FIDs suggest that image subtraction between the earliest TE (~10 μs) and a slightly longer TE (~25 μs) is a suitable choice for preserving the rapidly decaying component while suppressing the longer-lived ones ($T_2$ >100 μs) (*Figure 4*). The use of just two TEs to isolate collagen is advantageous for in vivo applications where minimal scan time is desirable. The subtraction yields an image that predominantly depicts collagen. This has been confirmed based on simulated acquisitions with tissue components according to a signal model replicating the FIDs (see Appendix 4). Increasing the TE interval will increase collagen sensitivity at the expense of specificity. The best trade-off between the two will vary depending on the features to be highlighted. Notably, selective collagen depiction by subtraction worked equally well with treated and untreated samples. This is important for the in vivo scenario where treatment is not an option.

Direct collagen imaging of the forearm in vivo yielded an anatomically insightful image (*Figure 5*). Dense, collagen-rich structures such as tendon and bone appear bright while the less dense, collagen-poorer muscle is largely suppressed. The detailed investigation on tendon and bone samples, in which collagen dominates as a contributor to the shortest $T_2$s, supports the interpretation of the in vivo data as reflecting collagen density with high specificity. Overall, image SNR was found to be remarkably good (~24.9 for cortical bone and ~16.6 for tendon – see Appendix 6) considering the extreme timescale of signal encoding, acquisition, and decay. For skin and muscle, which are also depicted in the collagen image, the origin of the short-$T_2$ signal is likely collagen with additional macromolecular contributions that remain to be clarified. Other anatomies (such as the knee and ankle) and tissues (such as dura and dentin) are collagen-rich and of interest for future in vivo studies with the same setup as used in this work. Direct collagen imaging is a promising method also for the study of fibrosis. Thus far, in vivo MR studies of fibrosis are limited to indirect methods, such as MR-elastography (*Venkatesh et al., 2013*) and MT (*Chang et al., 2024*). Direct collagen imaging in fibrosis is of clinical interest particularly in the torso, calling for high-performance whole-body gradients. Ongoing advances in gradient engineering are achieving adequate amplitudes (*Gudino and Littin, 2023*). However, full duty-cycle, as required for efficient collagen imaging, has yet to be achieved with whole-body, high-performance designs.

So far, conventional clinical MRI systems lack the capability of sustained high gradient strengths and rapid RF switching as required to image collagen directly. However, small-bore preclinical MRI systems often offer advanced gradient and RF capabilities, making them potentially more suitable for direct collagen imaging in ex-vivo samples as well as for in vivo small animal models. Nevertheless, gradient duty cycle limitations must still be taken into consideration. Another limitation for in vivo imaging can be the high RF power deposition of the high-bandwidth excitation required by the sequence. Although not the case in the presented example, this can effectively restrict the excitation flip angle and thus SNR, depending on the particular anatomy and RF coil (*Weiger and Pruessmann, 2019*; *Weiger et al., 2020*). In this respect, new developments for reducing the specific absorption rate are promising (*Baadsvik et al., 2025*).

In conclusion, MRI has the capacity to image collagen directly when performed at the timescale of $T_2$ on the order of 10 µs. This is possible with high-amplitude, full-duty-cycle gradient instrumentation, fast RF switching, and targeted acquisition protocols as derived here from initial characterization of the collagen signal. Direct collagen MRI has the potential to open a new field of research as well as clinical applications. It will be instrumental to quantitative studies on collagen-rich tissues in vivo, in which the macromolecular fractions have so far been determined only indirectly (*Jerban et al., 2019*). The ways in which pathology manifests in collagen images remain to be seen and could be investigated using studies on manipulated and pathological tissues. However, the ubiquity of collagen, its role in prevalent diseases, and the ability to image it with nuance and adequate SNR suggest significant diagnostic promise.

## Materials and methods
### Preparation of collagen-rich samples
Bovine Achilles tendon and femoral bone (~80% and~25–30% collagen type I by dry mass, respectively *Hudson et al., 2021*; *Sołtysiak et al., 2018*) were acquired from a local butcher. The specimens were cleaned of excess fat and muscle tissue. The bone marrow was removed, and a section of cortical bone was cut using a saw. Tendon and cortical bone samples were cut to approximately $10\times25$ mm$^2$. The samples were stored frozen before being thawed and brought to room temperature for the imaging experiments. The samples underwent a procedure to largely remove the sources of

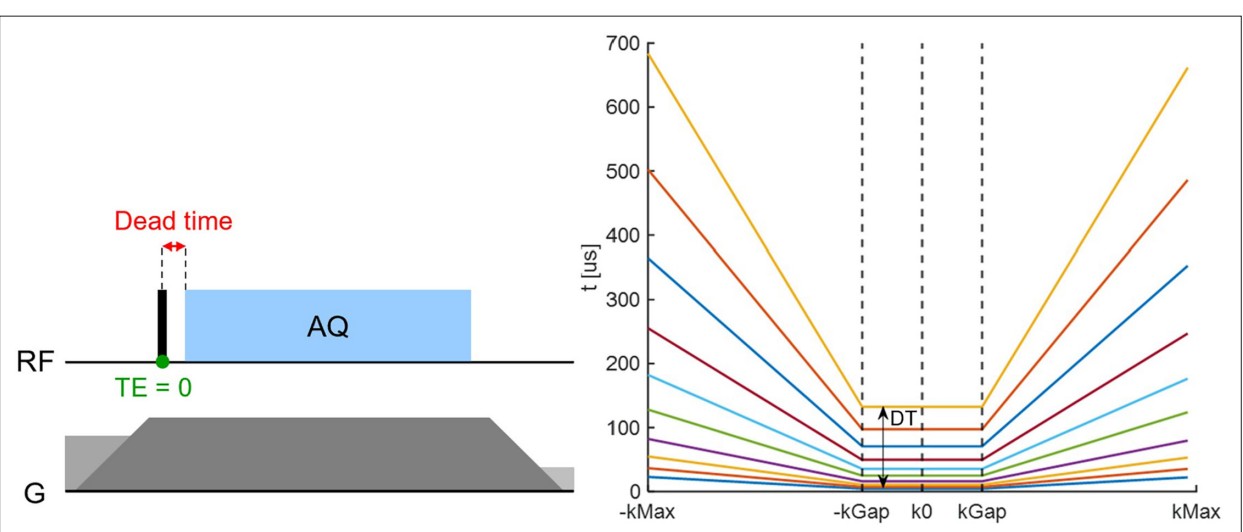

**Figure 6.** The short-$T_2$ PETRA protocol employed for direct collagen magnetic resonance imaging (MRI). (**a**) Basic zero echo time (ZTE) pulse sequence (adapted from *Weiger and Pruessmann, 2019*). After the radiofrequency (RF) dead time (DT), 3D radial encoding is performed to collect either ZTE data along radial spokes in k-space or single-point-imaging (SPI) data to fill the central k-space gap caused by the DT. (**b**) Corresponding t(k) plot, showing the time after excitation when each k-space data point is acquired. Inside the gap, the SPI data forms a plateau, whereas outside, the ZTE data forms a linear increase. To observe signal decay, the DT is increased in successive imaging experiments. The k-space gap (kGap = DT×G) is kept constant by reducing the gradient strength, leading to a steeper slope of t(k).

the water signal in the MR experiments (*Ma et al., 2016*). The procedure used $D_2O$ exchange for 4 days (replacing the $D_2O$ after 2 days) followed by freeze-drying the samples for 3 days. In this way, water molecules should either be removed from the sample or not provide [1]H signal. The MR experiments were performed on the same samples before and after the procedure and are referred to as 'untreated' and 'treated,' respectively. Care was taken to place the samples in the same orientation with respect to the main magnetic field to avoid signal changes due to orientation.

## NMR & MRI experiments

MR experiments were performed on a 3T Philips Achieva system (Philips Healthcare, the Netherlands) equipped with a custom gradient capable of reaching a strength of 220 mT/m at 100% duty cycle in a bore size of 33 cm (*Weiger et al., 2018*), and rapid transmit-receive switches (*Schildknecht et al., 2021*). Samples were studied using an RF loop coil of 40 mm diameter, and the in vivo experiment was performed using a quadrature birdcage coil of 100 mm diameter (*Weiger et al., 2026*). Both RF coils have been designed to largely avoid [1]H background signal from their materials.

FIDs were acquired on the tendon and bone samples. The purpose of acquiring the FIDs was to observe the overall signal behavior as well as to characterize tissue components for further experiments and simulations. The FID protocol was as follows: 2 µs block pulse, flip angle of 30°, repetition time (TR) of 30 ms, and 10,000 signal averages.

MR imaging was performed using a pulse sequence known as pointwise encoding time reduction with radial acquisition (PETRA) (*Grodzki et al., 2012*), which is a zero echo time-based technique (*Weiger and Pruessmann, 2023*) in which the central gap in k-space is acquired using single-point imaging (*Balcom et al., 1996*). The gap given in Nyquist intervals is defined as kGap = DT×BW, where DT is the dead time and BW is the imaging bandwidth (see *Figure 6*; *Baadsvik et al., 2024*). The sequence was performed on the tissue samples with DT ranging from 10 to 321 µs to observe the signal behavior with spatial localization. The bandwidth was adjusted such that the gap was constant for all acquired DTs. The effective TE values were assigned as TE = DT+Δ, where Δ accounts for averaging oversampled data over a Nyquist dwell 1/BW. This assignment was found to be appropriate by means of simulations (see Appendix 7).

In vivo experiments were performed in the forearm of a male volunteer. Written informed consent and consent to publish was obtained. The experiments were conducted in agreement with regional ethics regulations as approved by Swissmedic (approval ID 10000216). The arm was immobilized by means of inflatable cushions. Two TEs of 10.9 µs and 27.4 µs were chosen based on the results of the FID and multi-TE acquisition of the samples, as well as simulations. The imaging parameters are shown in *Table 1*.

## Data processing

Raw data was demodulated at the water frequency to prevent off-resonance blurring at lower BWs. At the associated larger TEs, the off-resonant collagen signal has already decayed sufficiently to consider its contribution negligible. Images were reconstructed by iterative k-space re-gridding (*Pruessmann et al., 2001*) and application of a hyperbolic secant windowing function (exponent 10, truncation factor 5) (*Baadsvik et al., 2024*; *Tesiram, 2010*) to remove Gibbs ringing effects

**Table 1.** Parameters for multi-echo time (TE) imaging of collagen-rich samples and in vivo human forearm. Abbreviations: TE, echo time, TR, repetition time, BW, image bandwidth.

| Protocol | Samples | In vivo forearm |
| --- | --- | --- |
| TE (µs) | 10.4–347.4 | 10.9, 27.4 |
| BW (kHz) | 598–18.7 | 1215, 486 |
| Excitation pulse | 2 µs block, 5.7° | 2 µs frequency-swept (*Schieban et al., 2015*), 2.7° |
| TR (ms) | 3.0 | 1.0 |
| Field of view (mm) | 64 | 130 |
| Nominal resolution (isotropic) (mm) | 1.0 | 0.98 |
| Scan time per image (m:s) | 02:00 | 09:38 |

from slower decaying signals. To observe the signal behavior of certain locations in the samples, ROIs were selected and the mean signal intensity was plotted as a function of TE. From these plots, two TEs were selected to perform image subtraction for preserving the shortest-living component while suppressing the longer-living components in the samples (*Li et al., 2012*; *Johnson et al., 2017*; *Szeverenyi and Carl, 2012*; *Rahmer et al., 2006*; *Ma et al., 2021*). The selected TEs also served as a guide when choosing imaging parameters for the in vivo application, where image subtraction was performed after registering the images. In the subtraction image, negative values were set to zero (*Lee et al., 2020*). For visualization purposes and SNR calculations, a twofold bi-cubic interpolation was applied, and the average over three neighboring slices was performed for the in vivo case.

## Acknowledgements

The authors would like to thank Prof. Matthias Ernst at the department of Chemistry and Applied Biosciences at ETH Zurich for sharing his expertise in solid-state NMR.

## Additional information

### Funding
No external funding was received for this work.

### Author contributions
Jason Daniel Van Schoor, Conceptualization, Data curation, Software, Formal analysis, Validation, Investigation, Visualization, Methodology, Writing – original draft, Project administration, Writing – review and editing; Markus Weiger, Conceptualization, Resources, Software, Formal analysis, Supervision, Validation, Investigation, Methodology, Project administration, Writing – review and editing; Emily Louise Baadsvik, Software, Methodology, Writing – review and editing; Klaas P Pruessmann, Conceptualization, Resources, Supervision, Funding acquisition, Validation, Project administration, Writing – review and editing

### Author ORCIDs
Jason Daniel Van Schoor ⓘ https://orcid.org/0009-0007-0159-3353
Markus Weiger ⓘ https://orcid.org/0000-0003-4758-3103
Emily Louise Baadsvik ⓘ https://orcid.org/0000-0001-5941-5532
Klaas P Pruessmann ⓘ https://orcid.org/0000-0003-0009-8362

### Ethics
Human subjects: Informed consent and consent to publish was obtained from the scan volunteer. The experiments were conducted in agreement with regional ethics regulations as approved by Swissmedic (approval ID 10000216).

Reviewer #1 (Public review): https://doi.org/10.7554/eLife.109799.3.sa1
Reviewer #2 (Public review): https://doi.org/10.7554/eLife.109799.3.sa2
Reviewer #3 (Public review): https://doi.org/10.7554/eLife.109799.3.sa3
Author response https://doi.org/10.7554/eLife.109799.3.sa4

## Additional files

### Supplementary files
MDAR checklist

### Data availability
Data from physcial experiments available: https://doi.org/10.5281/zenodo.15928442.

The following dataset was generated:

| Author(s) | Year | Dataset title | Dataset URL | Database and Identifier |
|---|---|---|---|---|
| van Schoor JD | 2025 | Direct MRI of Collagen | https://doi.org/10.5281/zenodo.15928442 | Zenodo, 10.5281/zenodo.15928442 |

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

# Appendix 1

## Estimation of contributions to proton signal

Contributions from different tissue components to the MR signal are estimated here through the approximation of the proton count associated with the principal proton-containing compounds found in tendon and cortical bone tissues. This analysis supports the attribution of the dominant, rapidly decaying signal component largely to protons on the collagen molecule.

The number of protons per gram is calculated as:

$$\frac{protons}{g} = \frac{N_a}{MW} \times \frac{protons}{molecule} \tag{1}$$

Where $N_a$ is Avogadro's number and $MW$ is the molecular weight in g/mol. The results are listed in **Appendix 1—table 1**.

To estimate the signal contributions for a specific tissue, the proton counts of the compounds are weighted by their percentage per mass in the tissue. For tendon, we assume 63% water and 27% collagen by mass according to **Taye et al., 2020**. The remaining 10% comprise elastin and other extracellular proteins which are not considered here. For cortical bone, we assume 8% water, 22% collagen, and 70% hydroxyapatite by mass taken from **Augat and Schorlemmer, 2006**.

By normalization, the estimated signal fractions are obtained as shown in **Appendix 1—table 2**. Protons from collagen contribute a significant proportion of the total MR signal in both tendon and cortical bone – 21% and 59%, respectively. Notably, hydroxyapatite has a low proton density and thus contributes little to the signal in cortical bone, making collagen the dominant contributor to the rapidly decaying signal.

**Appendix 1—table 1.** Dominant compounds in tendon and cortical bone with relevant stoichiometric parameters.
The collagen chain of Glycine-Proline-Hydroxyproline is used since it is the most commonly occurring chain in collagen type I (**Naomi et al., 2021**).

| Compound | Chemical formula | Protons/ molecule | Molecular weight (MW) [g/mol] | Protons/g [$10^{21}$] |
|---|---|---|---|---|
| Collagen chain of Glycine-Proline-Hydroxyproline | $C_{12}H_{19}N_3O_5$ | 19 | 285.3 | 40.1 |
| Water | $H_2O$ | 2 | 18.0 | 66.9 |
| Hydroxyapatite | $Ca_5 \left(PO_4\right)_3 OH$ | 1 | 502.3 | 1.2 |

**Appendix 1—table 2.** Estimated signal fractions for tendon and cortical bone tissues.

| Tissue | Water | Collagen | Hydroxyapatite |
|---|---|---|---|
| Tendon | 0.79 | 0.21 | 0 |
| Bone | 0.36 | 0.59 | 0.05 |

## Appendix 2

## FID signal representation and fitting

Previous studies have modelled FIDs as a sum of decay functions, which produce the established $T_2$s for collagen and bound-water protons. However, this approach does not capture the characteristic dipolar oscillation arising from dipolar coupling (*Mroue et al., 2015*). Quantum-filtered experiments isolate the collagen component as a doublet with splitting frequencies on the order of tens of kHz (*Eliav and Navon, 2002*; *Ong et al., 2012*). This splitting can be modelled using a cosine term leading to the following signal model:

$$S\left(t\right) = e^{i\varnothing}\left[\sum_j a_j e^{i\Delta\omega_j t}\cos\left(2\pi f_j t\right)e^{-\left[\frac{t}{T_{2,j}}\right]^{E_j}}\right]$$

(2)

$\varnothing$ represents a global phase shift, $a$ signifies a component amplitude, $\Delta\omega$ indicates off-resonance due to chemical shift, $f$ represents the splitting frequency of the net dipolar-coupling effect, $T_2$ denotes a transverse decay parameter, and $E$ represents an exponential coefficient. The exponent $E$ can be varied from 1 to 2 to hybridize the line shape associated with the decay to between Lorentzian (common for solution NMR) and Gaussian (common for solid-state NMR) (*Krasnosselskaia, 2012*). For non-coupling components, the values of $f$ and $E$ can be fixed at 0 and 1, respectively.

*Appendix 2—figure 1* presents the result of fitting the FID of untreated bone using the model of *Equation 2*. The resulting component parameters are shown in *Appendix 2—table 1*. Two of the components exhibit rapid Gaussian-like signal decay with dipolar coupling. Similarly, *Appendix 2—figure 2* presents the fitting for the treated tendon sample using three rapidly decaying components (two Gaussian-like with dipolar coupling effects) listed in *Appendix 2—table 2*. In both cases, the short-lived collagen components are the largest contributors to the signal.

Notably, the net observed coupling phenomena is well described using the mathematical model and proves useful in isolating signal constituents. However, there are limitations in using this technique. Fitting a sum of exponentials, without good priors, is known to be an ill-conditioned problem yielding many plausible solutions. For instance, changing the number of components and allowed $T_2$s and amplitude ranges will alter the result. It, therefore, is not appropriate to unambiguously assign signal components to distinct proton pools. Furthermore, the model does not account for effects such as orientation, which has been shown to influence the collagen-bound water signal (*Krasnosselskaia, 2012*). Nevertheless, the modelling method is sufficient to inform simulations to further probe the possibility of direct collagen imaging.

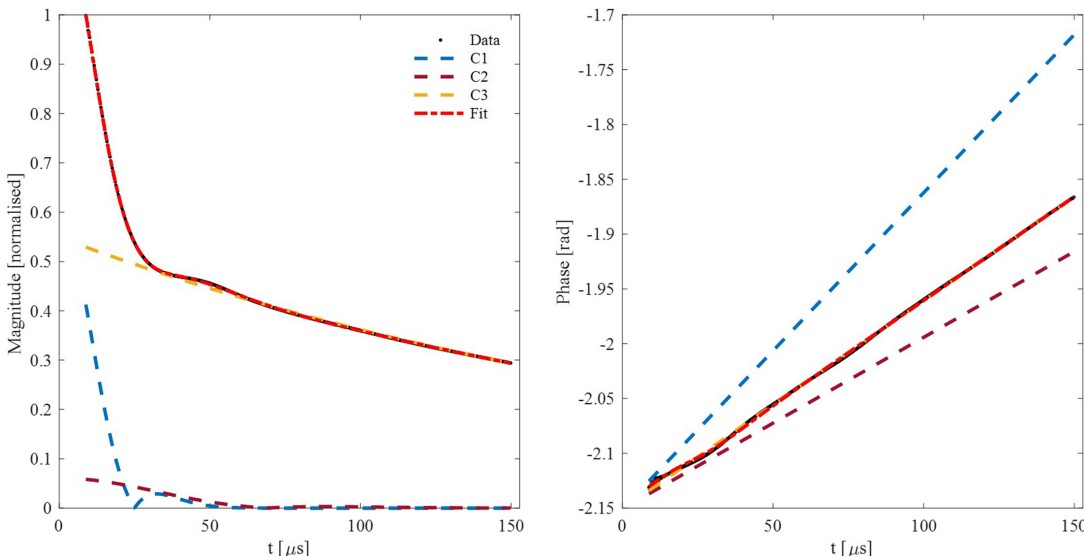

**Appendix 2—figure 1.** Free induction decay (FID) of untreated cortical bone sample with the accompanying model fit. The bump in the magnitude is attributed to an oscillation from dipolar coupling of the collagen protons.

The model fits the data well using three terms, of which two components (**C1 and C2**) are rapidly decaying dipolar coupling terms.

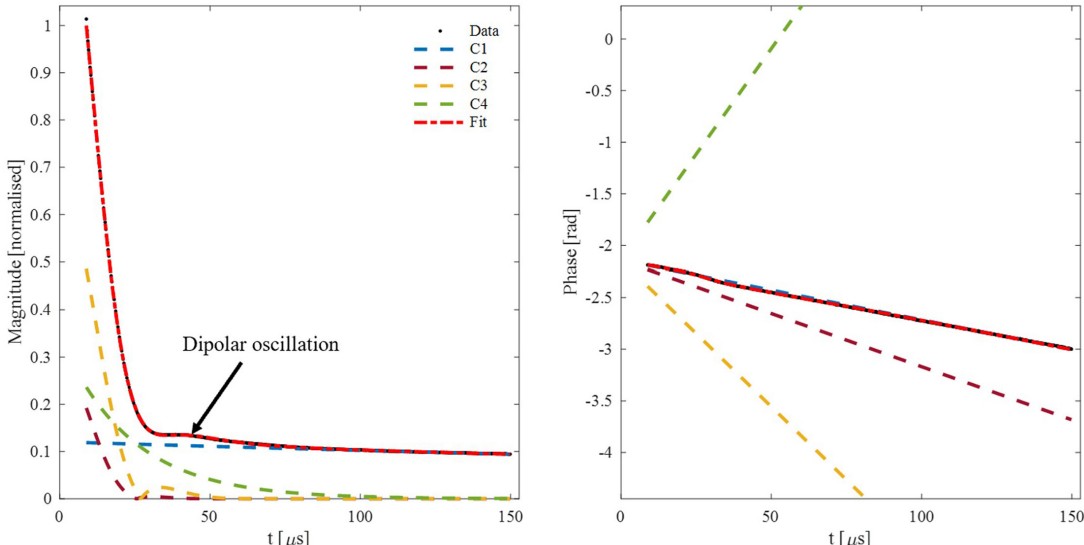

**Appendix 2—figure 2.** Free induction decay (FID) of treated tendon sample with the accompanying model fit. Again, the bump in the FID from dipolar coupling is preserved. The model fits using four terms with C2 and C3 being rapidly decaying components.

**Appendix 2—table 1.** Components from the fit in **Appendix 2—figure 1**.
The amplitudes are normalized according to the maximum signal observed in the free induction decay (FID). Notably, the short-lived components make the largest contribution. The values shown in bold were fixed during the fitting procedure.

| Component | Amplitude $a$ [norm.] | Off-resonance $\Delta\omega$ [kHz] | Splitting freq. $f$ [kHz] | Decay constant $T_2$ [µs] | Exponent $E$ |
|---|---|---|---|---|---|
| C1 | 0.497 | 0.459 | 10.000 | 20.7 | 1.800 |
| C2 | 0.050 | 0.249 | 3.633 | 60.0 | 2.000 |
| C3 | 0.453 | 0.302 | **0** | 239.8 | **1** |

**Appendix 2—table 2.** Components from fit in **Appendix 2—figure 2**.
The amplitudes are normalized according to the maximum signal observed in the free induction decay (FID). Here, the short-lived components dominate. The values shown in bold were fixed during the fitting procedure.

| Component | Amplitude $a$ [norm.] | Off-resonance $\Delta\omega$ [kHz] | Splitting freq. $f$ [kHz] | Decay constant $T_2$ [µs] | Exponent $E$ |
|---|---|---|---|---|---|
| C1 | 0.083 | −0.936 | **0** | 594.0 | **1** |
| C2 | 0.227 | −1.630 | 9.397 | 17.3 | 2.000 |
| C3 | 0.517 | −3.700 | 9.942 | 20.9 | 1.848 |
| C4 | 0.172 | 7.00 | **0** | 24.9 | **1** |

## Appendix 3

## Calculation of T$_2$ blurring

As is well-known in MRI, the time spent on encoding and acquiring signal with respect to T$_2^*$ or T$_2$ decay impacts both SNR and effective resolution. This aspect is particularly important when imaging tissues with very short T$_2$ (*Froidevaux et al., 2020*; *Rahmer et al., 2006*), where longer encoding times lead to increased T$_2$ blurring, corresponding to a reduction in effective resolution. For a desired nominal resolution, a large gradient strength enables a reduction in encoding time and is, therefore, key to contain resolution loss in short-T$_2$ MRI.

Here, we analyze the resolution of the collagen signal using the method presented for the PETRA pulse sequence by *Froidevaux et al., 2020*. This method assumes an exponentially decaying signal rather than the Gaussian-like decay for collagen modeled in Appendix 2. Therefore, we first derive a corresponding $T_2$ of an exponential decay from the fitted decay, $T_2^E$ and exponent, $E$, which are used to describe the Gaussian-like decay:

$$e^{-\frac{t}{T_2}} = e^{-\left[\frac{t}{T_2^E}\right]^E} \tag{3}$$

$$T_2 = T_2^E t^{1-E} \tag{4}$$

*Equation 4* is plotted in *Appendix 3—figure 1* using the parameters acquired from the modelled bone FID (*Appendix 2—table 1*). One observes that the equivalent exponential $T_2$ decreases with increasing time. Here, we choose $t = 40$ μs, which is late in the encoding time of the sequence with a dead time of 10 μs and, therefore, represents a worst-case scenario of T$_2$ blurring. t=40 μs is also the time at which the collagen signal appears to have decayed in the FIDs. The derived $T_2$ for collagen signal in bone is, therefore, taken to be 12 μs.

Calculations of effective resolution were conducted for the protocol with the shortest TE in *Table 1* and the parameters summarized in *Appendix 3—table 1*. The associated point-spread-functions (PSFs) without and with T$_2$ decay are shown in *Appendix 3—figure 2*. The full-width-at-half-maximum (FWHM) of the PSF is interpreted as the effective resolution. Without decay, this amounts to 1.6 mm, which is an increase of 60% with respect to the nominal resolution of 1 mm due to the width of the main lobe of the intrinsic, sinc-shaped PSF. For collagen, a FWHM of 2.2 mm is obtained, which is an increase of 38% with respect to the ideal case due to T$_2$ blurring. This calculation shows that whilst rapidly decaying signals are spatially resolvable, a significant amount of T$_2$ blurring is to be expected.

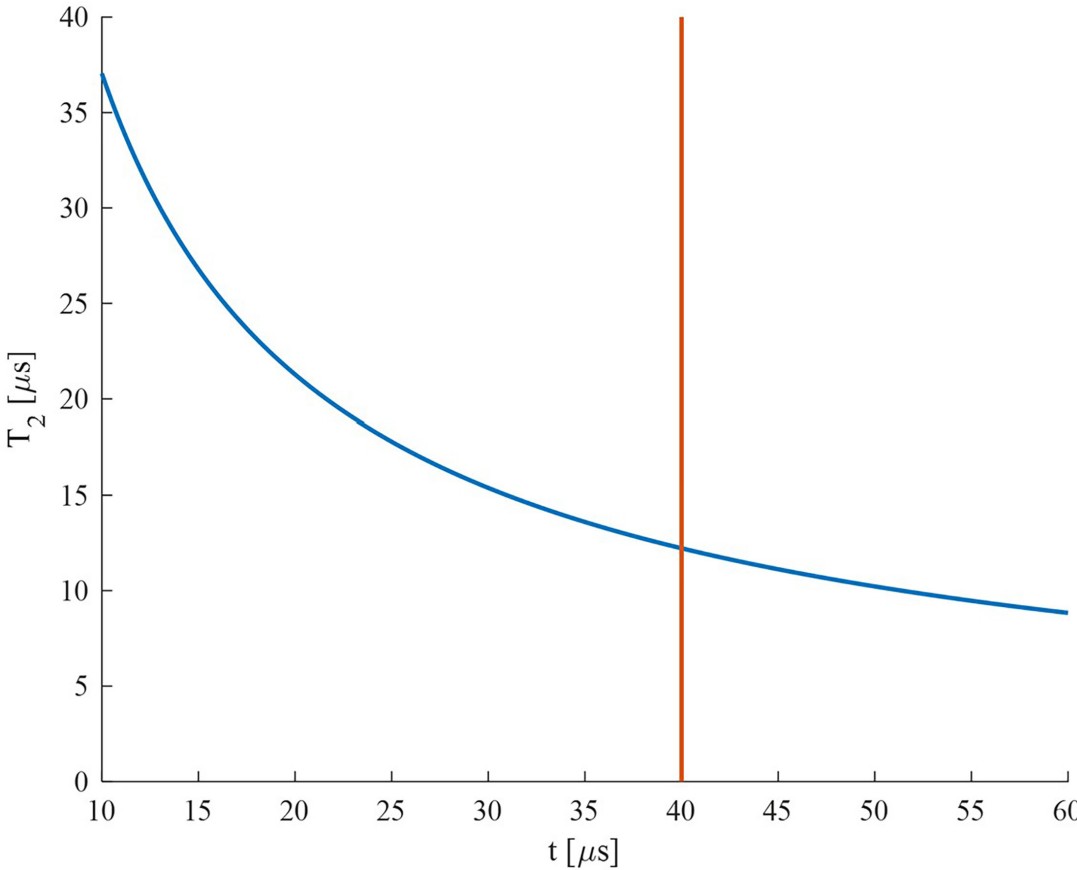

**Appendix 3—figure 1.** Exponential $T_2$ as a function of time $t$ according to **Equation 4** for Gaussian-like decay for parameters $T_2^E$ = 20.7 μs and $E$ = 1.8 (the fastest decaying component in **Appendix 2—table 1**). One observes that the equivalent exponential $T_2$ decreases with increasing time. At the indicated $t$ = 40 μs, $T_2$ is approximately 12 μs.

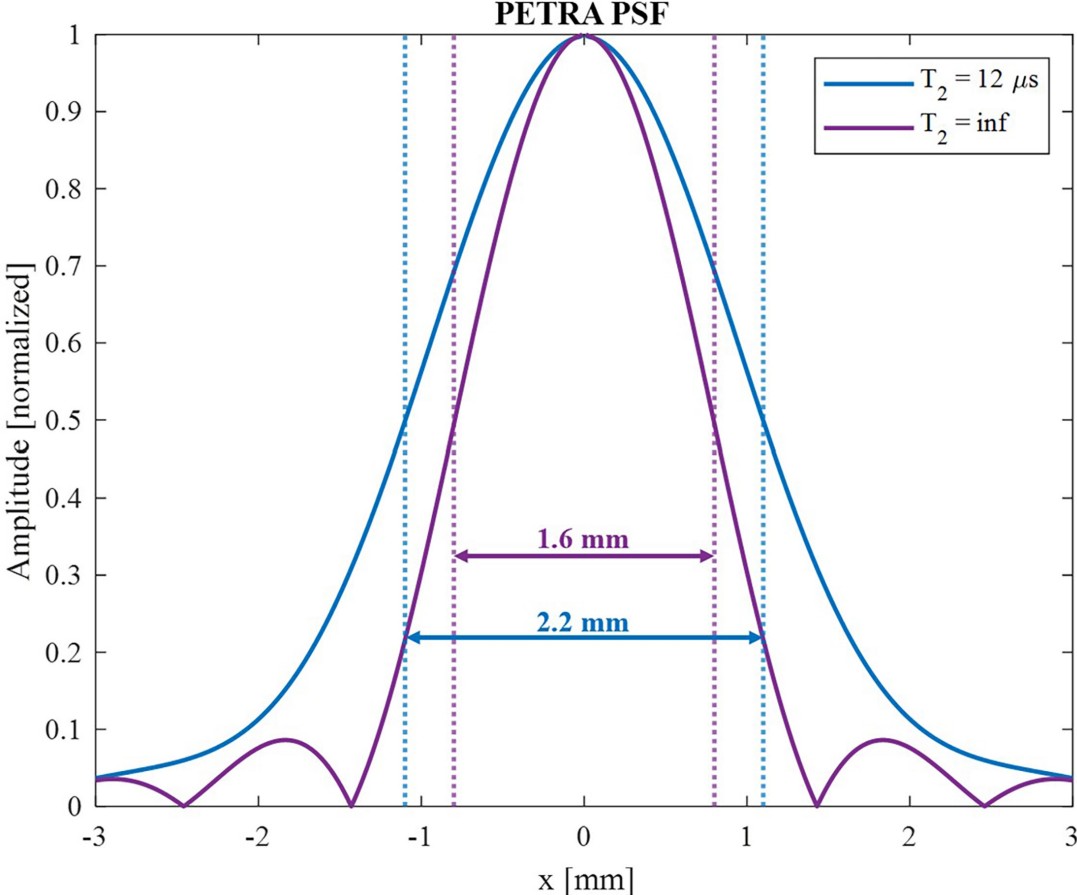

**Appendix 3—figure 2.** Point-spread-function (PSF) of pointwise encoding time reduction with radial acquisition (PETRA) pulse sequence for collagen signal with exponential $T_2$ decay of 12 µs, compared to that of a non-decaying signal, using imaging parameters listed in *Appendix 2—table 1*.

**Appendix 3—table 1.** Parameters used for calculation of $T_2$ blurring.

| $T_2$ [µs] | Gradient Strength [mT/m] | Dead time DT [µs] | Nominal resolution [mm] |
|---|---|---|---|
| 12 | 220 | 10 | 1 |

## Appendix 4

### Simulation of imaging and image subtraction

To investigate the specificity of image subtraction, i.e., potential contamination by residual water signal in the difference image, simulations were performed. Four spherical objects with distinct $T_2$ pools were simulated, where three spheres represent individual components and the fourth sphere represents the sum of all components. The same multi-TE PETRA sequence as in the imaging experiments was applied.

Appendix 4—figure 1 presents the simulation results. The collagen component is visible at TE = 10 μs and no longer visible at TE = 25 μs. The signal intensity of the bound and bulk water components appears unchanged between the two echoes. Subtracting the two images leaves only the collagen signal component. Merely a very weak residual signal from the bound-water component is observed. Taking the mean image intensities over ROIs in the spheres and plotting them as a function of TE reconstructs the FIDs of the individual and the combined tissue components. It is evident that in the early TE data, the rapidly decaying collagen component is captured. Intensity profiles of the simulated spheres were merged for display to show the effects observed in the images more quantitatively. The effects of $T_2$ blurring in the collagen sphere are obvious. In contrast, the bound-water sphere does not exhibit blurring, in particular not its contamination to the difference image. This contamination adds to the collagen signal in the combined-signal sphere, resulting in an apparently higher resolution of collagen in the difference image.

The simulations show that the PETRA pulse sequence successfully captures and spatially resolves the rapidly decaying collagen component, and that the collagen signal no longer contributes to images after TE = 50 μs. This would suggest that future investigations that seek only to study direct collagen signal could become more time efficient by excluding longer echo times. Image subtraction of early echo times convey that primarily the collagen signal component remains, with a small residual contamination from bound-water signal (signal with $T_2$s on the order of 100s of microseconds).

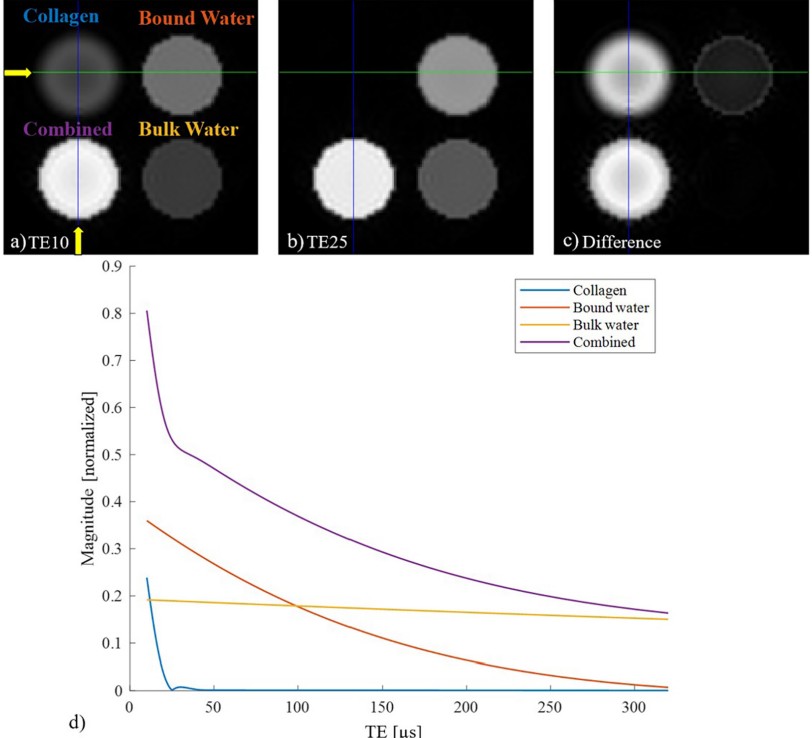

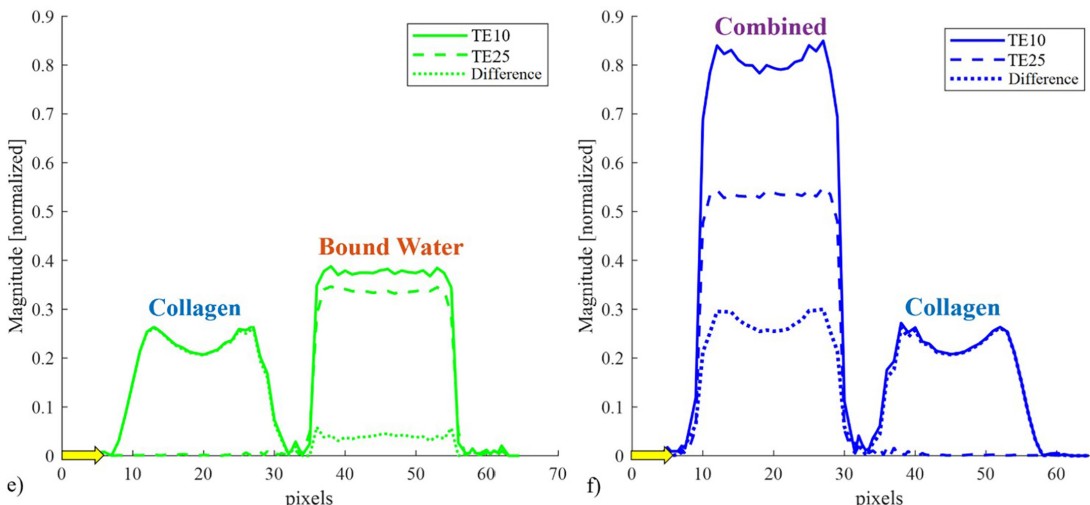

**Appendix 4—figure 1.** Imaging simulation of four spheres using collagen, bound-water, and bulk-water signal characteristics. (**i**) A 'Collagen' component with dipolar coupling (T2=10 µs, $f$=10 kHz, E=1, $\Delta\omega$=0.5 kHz), (**ii**) a 'Bound water' component (T2=100 µs, $f$=0.7 kHz, E=1), (**iii**) a 'Bulk water' component (T2=1.3 ms), and (**iv**) all components combined (adding all components into a single sphere). The spheres were simulated with a pointwise encoding time reduction with radial acquisition (PETRA) sequence with echo time (TE) ranging from 10 to 320 µs. The displayed images are scaled with respect to their own maximum intensity. (**a**) Simulated image at TE = 10 µs. The collagen sphere shows reduced signal and significant blurring. (**b**) Simulated image at TE = 25 µs. The collagen sphere has vanished while the other spheres persist. (**c**) Difference image of TE = 10 µs and TE = 25 µs. Only the collagen component remains. In addition, a residual ring of the intermediate component is observed, indicating a small signal contamination of ~4%. This ring also appears in the combined sphere, making it slightly sharper than the pure collagen sphere. (**d**) Plotted mean signal intensities over the region of interests (ROIs) drawn over the different spheres as a function of TE. The behavior of the underlying free induction decay (FID) signal is reproduced, including the rapid initial signal decay as captured by the shortest TEs. (**e**) and (**f**) Intensity profiles

*Appendix 4—figure 1 continued on next page*

*Appendix 4—figure 1 continued*

along spheres, illustrating effects of T2 blurring. The profiles show that the collagen sphere is notably more blurred than the bound-water and combined spheres. Furthermore, one observes that the contamination from the bound water adds to the difference profile of the combined signal sphere, resulting in a sharper-looking image. This explains why the T2 blurring from the collagen-only sphere appears more apparent.

# Appendix 5

## Multi-TE analysis of untreated tendon

To further analyze the peculiar contrast in the subtraction image (see *Figure 4* of main text) of the untreated tendon sample, the full signal decay is investigated. ROIs are drawn on two different regions of the sample and the mean signal intensity is plotted as a function of TE. The locations of the ROIs are informed by the bright band seen in the difference image shown in *Appendix 5—figure 1d*.

*Appendix 5—figure 1* shows that the initial signal amplitude differs between the two regions, the initial decay is stronger in the bright region, and their signal behavior further changes for TEs above 150 μs (the bound-water imaging interval). These observations indicate the presence of different tissue structures in the sample that – by means of the employed imaging approach – are observable at both the collagen and bound-water timescales. However, the exact mechanism behind this change of signal behavior is uncertain and would need a detailed analysis of the tissue. Notably, the same contrast is not visible in the subtraction of the treated sample, indicating that the treatment leads to changes in the tissue affecting the rapidly decaying signal. Overall, the results show that the direct collagen imaging technique is sensitive to changes in underlying tissue structure, which demonstrates its potential for in vivo MSK diagnostics.

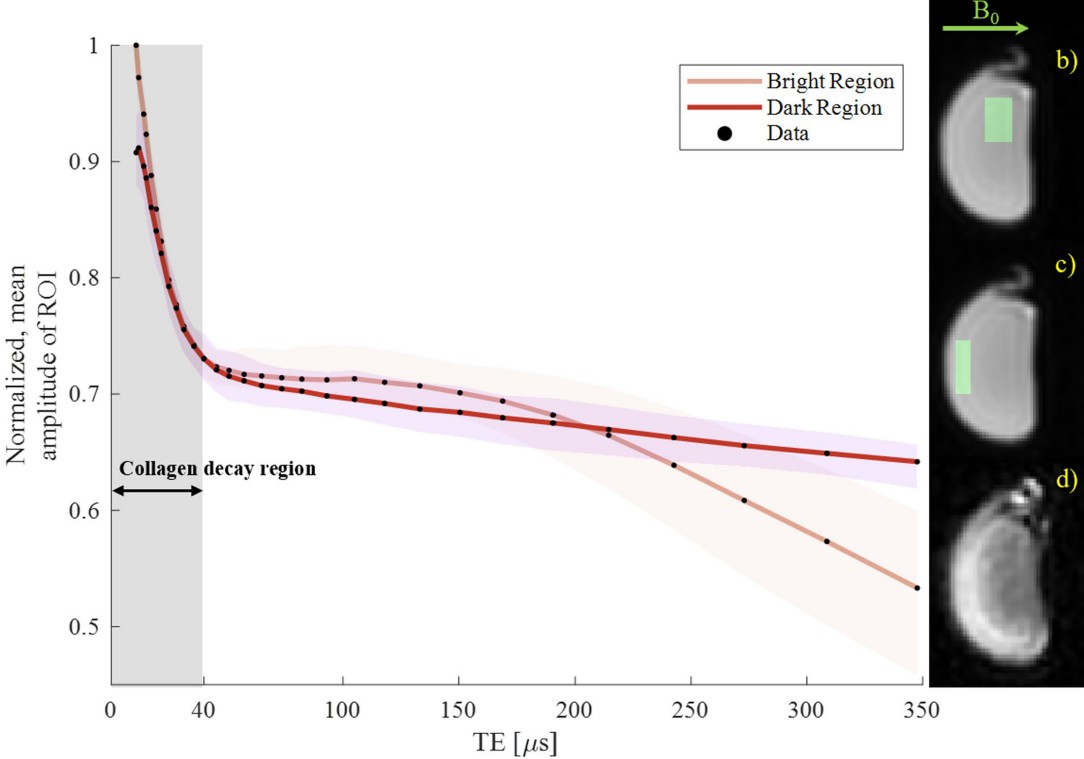

**Appendix 5—figure 1.** Mean signal intensity as a function of echo time (TE) over regions of interest (ROIs) in different parts of the untreated tendon sample. The signal in the bright region of the difference image (**d**) (ROI shown in panel (**c**) for the TE = 10.4 μs image) starts at a larger amplitude, initially decays faster, and shows a change in bound-water behavior after ~150 μs as compared to the darker region (ROI shown in panel (**b**) for the TE = 10.4 μs image). This change in signal characteristics is hypothesized to be a result of variation in tissue structure. The shaded areas around the lines indicate the 90% central ranges of the averaged data points.

## Appendix 6

### Determination of SNR for the in vivo collagen image

To determine the SNR of the in vivo collagen subtraction image of the forearm, circular ROIs were drawn over the cortical bone and tendon structures, as well as over areas of pure noise. The SNR was then calculated in the magnitude image as the average intensity of the voxels in the structure $\bar{S}$, divided by the standard deviation of the noise $std\,(N)$, i.e., $SNR = \bar{S}/std\,(N)$. Before calculating the SNR, as for the visualization, three neighboring slices were averaged. The results are shown in *Appendix 6—figure 1*.

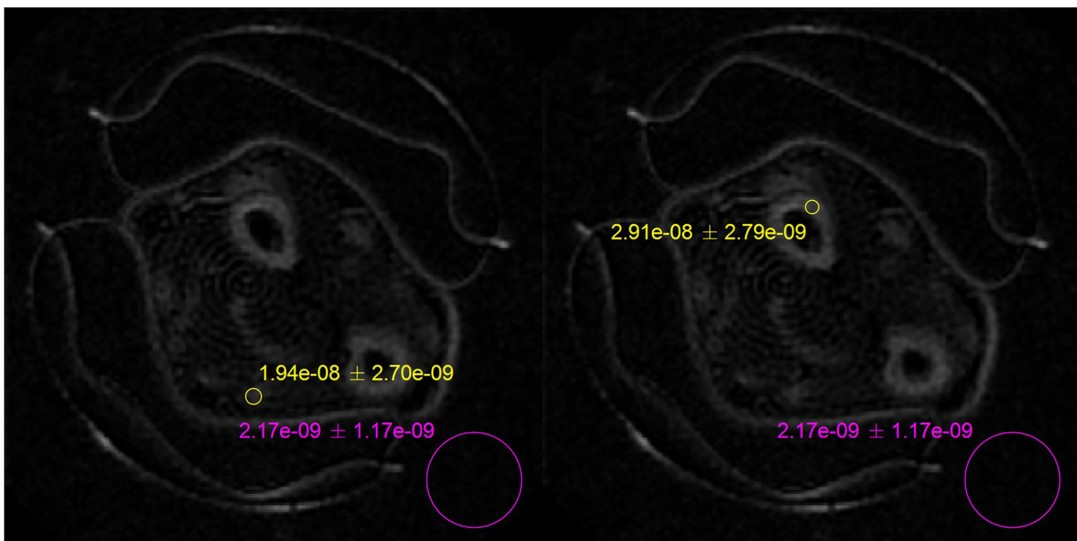

**Appendix 6—figure 1.** Region of interests (ROIs) drawn on collagen image of in vivo forearm for tendon (left) and cortical bone (right) structures. Anatomical ROIs are shown in yellow, and the noise ROIs are shown in pink. The signal-to-noise ratio (SNR) is calculated to be 24.9 and 16.6, respectively, including the three-slice averaging used for visualization.

## Appendix 7

### Confirmation of TE assignment

To assign a representative TE for an image with a given dead time (DT, the time between RF excitation and acquisition) for the PETRA pulse sequence, simulations of a single point imaging (SPI) acquisition with the same four spheres as described in *Section II* were performed (*Grodzki et al., 2012*; *Emid and Creyghton, 1985*). An SPI acquisition collects each k-space point with the same time delay after the RF excitation, and therefore all points have the same $T_2$ weighting. For SPI acquisitions, we assign TE = DT + $\Delta$, where $\Delta = \left(\frac{1}{2} - \frac{1}{2\,ov}\right)\frac{1}{BW}$ accounts for averaging data, which were oversampled with a factor $ov$, over a Nyquist dwell $1/BW$. For direct collagen MRI, pure SPI is not practically implementable due to the high gradient demands for larger k-space values. However, simulations allow one to compare the differences in signal magnitude between the acquisitions and then determine a correction factor for the DT of the PETRA pulse sequence such that a representable TE can be assigned (*Baadsvik et al., 2024*).

The results of simulating the SPI acquisition, as well as the difference in signal magnitude to the PETRA acquisition (from simulations performed and displayed in *Appendix 4—figure 1*) are shown in *Appendix 7—figure 1*. The SPI signal magnitudes prove to be very similar to those of the PETRA simulations with a very small difference in signal. For this reason, one may assign the TE for PETRA to be:

$$TE_{PETRA} \cong TE_{SPI} \tag{5}$$

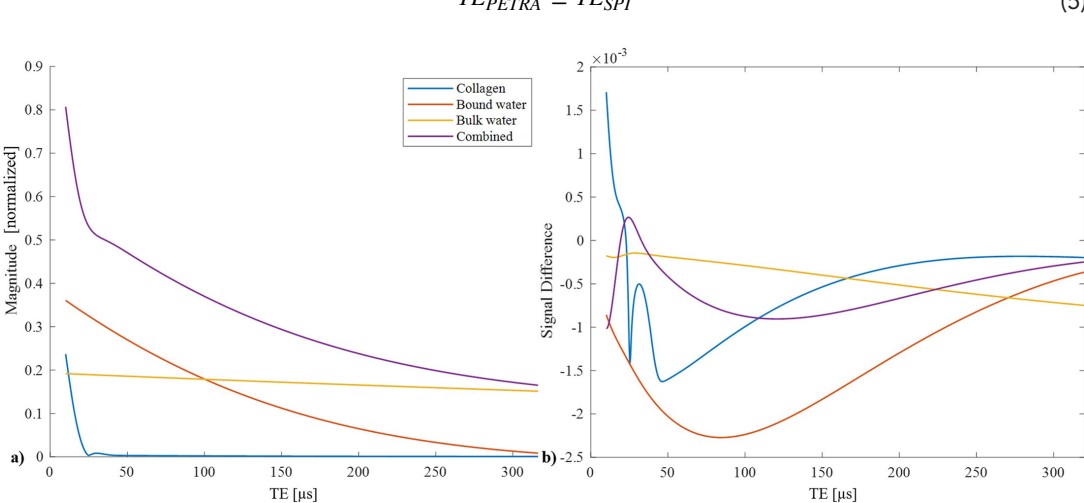

**Appendix 7—figure 1.** Comparison of echo times (TEs) between pointwise encoding time reduction with radial acquisition (PETRA) and single point imaging (SPI) acquisitions. (**a**) Region of interest (ROI) plot for simulated SPI acquisition using the same four spheres presented in *Appendix 4—figure 1a*. The simulated SPI acquisition yields uniform $T_2$ weighting to all points in k-space and its TE is assigned as TE = DT + $\Delta$. (**b**) Difference between signals of different components in the SPI and PETRA acquisitions. The signal difference is small, and this supports the same assignment of TE for the PETRA pulse sequences as for the SPI pulse sequence.

