## [Editor Report · eLife Assessment]

This **fundamental** work substantially advances our understanding of a major research question: whether collagen can be directly imaged with MRI. The evidence supporting the conclusion is **compelling**, with methods, data, and analyses that are more rigorous than those currently considered state-of-the-art. The work will be of high interest to MR physicists and clinicians, as collagen is the most abundant protein in the human body and plays an essential role in health.

---

## [Referee Report · Reviewer #1 (Public review)]

Summary:

The aim of this work is to directly image collagen in tissue using a new MRI method with positive contrast. The work presents a new MRI method that allows very short, powerful radio frequency (RF) pulses and very short switching times between transmission and reception of radio frequency signals.

Strengths:

The experiments with and without removal of 1H hydrogen, which is not firmly bound to collagen, on tissue samples from tendons and bones are very well suited to prove the detection of direct hydrogen signals from collagen. The new method has great potential value in medicine, as it allows for better investigation of ageing processes and many degenerative diseases in which functional tissue is replaced by connective tissue (collagen).

Comments on revisions:

All points of criticism in the reviews were answered very well and led to further improvement of the article.

---

## [Referee Report · Reviewer #2 (Public review)]

Summary:

This work presents direct magnetic resonance imaging (MRI) of collagen, which is not possible with conventional MRI or other tomographic imaging modalities.

Strengths:

The experimental work is impressive, and the presentation of results is clear and convincing.

---

## [Referee Report · Reviewer #3 (Public review)]

The paper is well written and well presented. The topic is important, and its significance is explained succinctly and accurately. I am only capable of reviewing the clinical aspects of this work which is very largely technical in nature. Several clinical points are worth considering:

(1) Tendons typically display large magic angle effects as a result of their highly ordered collagen structure (cortical bone much less so) and so it would have been of interest to know what orientation the tendons had to B 0 (in vitro and in vivo). This could affect the signal level at the longer echo time and thus the signal on the subtracted images.

(2) The in vivo transverse image looks about mid-forearm where tendons are not prominent. A transverse image of the lower forearm where there is an abundance of tendons might have been preferable.

(3) The in vivo images show the interosseous membrane as high signal on both the shorter and longer TE images. The structure contains ordered collagen with fibres at different oblique angles to the radius and ulnar and thus potentially to B 0. Collagen fibres may have been at an orientation towards the magic angle and this may account for the high signal on the longer TE image, and the low signal on the subtracted image.

(4) Some of the signals attributed to muscle may be from an attachment of the muscle to aponeurosis.

(5) There is significant collagen in subcutaneous tissues so the designation "skin" may more correctly be "skin and subcutaneous tissue".

(6) Cortical bone is very heterogeneous with boundaries between hard bone and soft tissue with significant susceptibility differences between the two across a small distance. This might be another mechanism for ultrashort T 2 * tissue values in addition to the presence of collagen. The two effects might be distinguished by also including a longer TE spin echo acquisition.

Solid cortical bone may also have an ultrashort T 2 * in its own right.

(7) It may be worth noting that in disease T 2 * may be increased. As a result, the subtraction image may make abnormal tissue less obvious than normal tissue. Magic angle effects may also produce this appearance.

(8) It may be worth distinguishing fibrous connective tissue (loose or dense) which may be normal or abnormal, from fibrosis which is abnormal accumulation of fibrous connective tissue in damaged tissue. Fibrosis typically has a longer T 2 initially and decreases its T 2 * over time. In places, the context suggests that fibrous connective tissue may be more appropriate than fibrosis.

Overall, the paper appears very well constructed and describes thoughtful and important work.

Comments on revisions:

The responses to my criticisms are well thought out and are fine as far as I am concerned.

I suggest in Figure 5 line 6 changing "trabecular bone" to "trabecular bone marrow".

---

## [Author Response]

The following is the authors’ response to the original reviews

**Public Reviews:**

**Reviewer #1 (Public review):**
Summary:The aim of this work is to directly image collagen in tissue using a new MRI method with positive contrast. The work presents a new MRI method that allows very short, powerful radio frequency (RF) pulses and very short switching times between transmission and reception of radio frequency signals.Strengths:The experiments with and without the removal of 1H hydrogen, which is not firmly bound to collagen, on tissue samples from tendons and bones, are very well suited to prove the detection of direct hydrogen signals from collagen. The new method has great potential value in medicine, as it allows for better investigation of ageing processes and many degenerative diseases in which functional tissue is replaced by connective tissue (collagen).Weaknesses:It is clear that, due to the relatively long time intervals between RF excitation and signal readout, standard hardware in whole-body MRI systems can only be used to examine surrounding water and not hydrogen bound to collagen molecules.

We agree that this is a regrettable situation (see also Discussion section). We are hoping that current and future efforts of MRI manufacturers towards improved hardware will eventually enable the technique for broader application.

**Reviewer #2 (Public review):**
Summary:This work presents direct magnetic resonance imaging (MRI) of collagen, which is not possible with conventional MRI or other tomographic imaging modalities.Strengths:The experimental work is impressive, and the presentation of results is clear and convincing. Through a series of thoughtfully prepared experiments, I found the evidence that the images reflect direct measurements of collagen to be highly compelling.

Due to the technical demands, direct collagen imaging is unlikely to become widespread for routine clinical work, at least not anytime soon. That said, this work is nonetheless transformative and will likely be highly significant for research and perhaps clinical trials.

**Reviewer #3 (Public review):**
The paper is well written and well presented. The topic is important, and its significance is explained succinctly and accurately. I am only capable of reviewing the clinical aspects of this work, which is very largely technical in nature. Several clinical points are worth considering:(1) Tendons typically display large magic angle effects as a result of their highly ordered collagen structure (cortical bone much less so), and so it would have been of interest to know what orientation the tendons had to B 0 (in vitro and in vivo). This could affect the signal level at the longer echo time and thus the signal on the subtracted images.

We have added arrows in the images showing the direction of the main magnetic field. For the in vivo case, the subject lay in the superman position, with B0 pointing from the hand towards the shoulder.

(2) The in vivo transverse image looks about mid-forearm, where tendons are not prominent. A transverse image of the lower forearm, where there is an abundance of tendons, might have been preferable.

We have added a distal view of the forearm, where more tendon structures are observed.

(3) The in vivo images show the interosseous membrane as a high signal on both the shorter and longer TE images. The structure contains ordered collagen with fibres at different oblique angles to the radius and ulnar, and thus potentially to B 0. Collagen fibres may have been at an orientation towards the magic angle, and this may account for the high signal on the longer TE image and the low signal on the subtracted image.

This is certainly an interesting take. While the magic angle effect is well established for collagen bound water, the orientation effects on the macromolecular collagen signal are still to be investigated. Our initial experiences so far suggest that the direct collagen signal is not as sensitive to orientation as the bound water.

Regarding the described observation for the interosseous membrane, we expect the high signal coming from collagen-bound water (yet not quite at the magic angle), which hardly decays between the two TEs, as their difference is small as compared to the T2* of this signal. Hence, this signal is removed in the subtraction image, and only the macromolecular collagen signal remains, which appears to be very low. Working with samples of the interosseus membrane may provide further insights into why this is the case.

(4) Some of the signals attributed to the muscle may be from an attachment of the muscle to the aponeurosis.

We have added the aponeurosis as a possible signal contributor in the muscle tissue.

(5) There is significant collagen in subcutaneous tissues, so the designation "skin" may more correctly be "skin and subcutaneous tissue".

We have updated the label accordingly.

(6) Cortical bone is very heterogeneous, with boundaries between hard bone and soft tissue with significant susceptibility differences between the two across a small distance. This might be another mechanism for ultrashort T 2 * tissue values in addition to the presence of collagen. The two effects might be distinguished by also including a longer TE spin echo acquisition.

Solid cortical bone may also have an ultrashort T 2 * in its own right.

The described effect is clearly of importance for bone water but plays a negligible effect for the macromolecular signal. We would like to support this by a brief, coarse estimation. 𝑇_2_* can be approximated by 1/𝑇_2_* = 1/𝑇_2_ + 1⁄𝑇_2_′, where 1⁄𝑇_2_′ \ = 𝛾∆𝐵 = 𝛾∆𝜒𝐵_0_ (Ref. 1).

The susceptibilty difference reported for the interface between bone and water is ∆𝜒 = 2.5 ppm (Refs. 2 and 3), which at 3T leads to a 𝑇_2_′ ≈ 3000 𝜇𝑠. From our recorded FIDs, we use a 𝑇_2_* of 10 μs and thus obtain 𝑇_2_ \ = 10.03 𝜇𝑠.

As can be seen, the change in the transverse relaxation constant due to susceptibility is negligible compared to the intrinsic decay of the macromolecular collagen signal. Notably, this is not the case for the pore water signal where T_2_s are on the order of milliseconds (Ref. 2).

A footnote was added in the Introduction section regarding this topic.

(7) It may be worth noting that in disease T 2 * may be increased. As a result, the subtraction image may make abnormal tissue less obvious than normal tissue. Magic angle effects may also produce this appearance.

This is an important point regarding image interpretation. For this reason, it is advantageous that also the original anatomical images prior to subtraction are available, which will show such effects. They can be used in conjuction with the collagen-specific image to provide further insights regarding tissue disease. Increased T_2_* of diseased tissue has so far been reported for the bound water components due to a reduction of dipolar interactions between bound water and collagen (Ref. 4). A potential related change in T_2_ for the macromolecular collagen component itself is certainly of interest and an avenue to explore in future work.

(8) It may be worth distinguishing fibrous connective tissue (loose or dense), which may be normal or abnormal, from fibrosis, which is an abnormal accumulation of fibrous connective tissue in damaged tissue. Fibrosis typically has a longer T 2 initially and decreases its T 2 * over time. In places, the context suggests that fibrous connective tissue may be more appropriate than fibrosis.

We are aware of this important distinction. We therefore checked the manuscript for references to fibrosis, making sure that the meaning is as intended.

Overall, the paper appears very well constructed and describes thoughtful and important work.
**Recommendations for the authors:**

**Reviewer #1 (Recommendations for the authors):**
(1) It should be stated that various methods with very short echo times (e.g. SWIFT by Garwood et al.) have been described in the past. This work shows for the first time that direct signals from collagen and be systematically detected in tissue samples.

We have expanded a sentence in the introduction and reference selected publications studying short-T_2_ water signal in collagen, including SWIFT.

(2) It should be noted that the 1H atoms bound to collagen are located at different sites (at different amino acids of the protein) of the molecule and have different frequencies, and that further signal analyses are of interest.

We have included additional information regarding distinct resonances of proton-binding sites of collagen in the introduction. The discrete observation of such signals requires advanced NMR methodology such as magic-angle spinning and RF decoupling, which is not a suitable approach for in vivo MRI. Without such methods, the broad lineshapes overlap strongly and are rather observed as a single decaying exponential with the dipolar oscillation as we observe in the FIDs.

(3) Is it certain that the bump at 30 microseconds comes from 'dipolar coupling'? Is the development time probably too short for chemical shift-induced interference or J-coupling effects?

30 microseconds is an extremely short interval to accumulate phase and requires large resonance offsets to observe significant changes. To investigate the nature of the bump, we also collected data on a Bruker 7T NMR spectrometer (see Author response image 1). Overall the same signal characteristics are observed as with 3T. In particular, the position of the bump is the same, excluding chemical shift as as source. However, with the higher field strength, chemical shift becomes significant for the signal phase, as observed by the change in the phase behavior at 50 microseconds, when the collagen component has decayed.

While J-coupling is independent of field strength, the typical ranges are single-digit to tens of Hertz. In contrast, dipolar coupling interacts on the order of thousands of Hertz, which coincides with the values extracted from our signal model.

To clarify this point, we extended the respective sentence in the Results section.

**Author response image 1. sa4fig1:** 

(4) It should be noted that short RF pulses have a relatively high energy content, and whether there are any particular stresses on patients during the examination (SAR, nerve stimulation?).

SAR is an important issue in ZTE MRI. Since imaging bandwidths are large and excitation is performed with the imaging gradient being on, broadband pulses are necessary. Hence, significant RF deposition occurs and in vivo the flip angle can often not be optimized for the maximum signal, but will be limited by the SAR limit. We have added an explanation in the Discussion section.

Peripheral nerve stimulation is generated by rapid switching of strong gradients. However, ZTE sequences are usually operated without switching gradients on and off, but with only minor adjustments of the gradient direction between TR intervals. Therefore, PNS is not a relevant issue.

(5) In the Results section, Part B, 'substantial signal intensity' should be written instead of 'substantial image intensity'.

We have changed this as suggested.

References

(1) Chavhan GB, Babyn PS, Thomas B, Shroff MM, Haacke EM. Principles, techniques, and applications of T2*-based MR imaging and its special applications. Radiographics. 2009 Sep-Oct;29(5):1433-49. doi: 10.1148/rg.295095034. PMID: 19755604; PMCID: PMC2799958.

(2) Seifert, AC, Wehrli, SL, and Wehrli, FW (2015), Bi-component *T*_2_* analysis of bound and pore bone water fractions fails at high field strengths. NMR Biomed., 28, 861– 872. doi: 10.1002/nbm.3305.

(3) Hopkins JA, Wehrli FW. Magnetic susceptibility measurement of insoluble solids by NMR: magnetic susceptibility of bone. Magn Reson Med. 1997 Apr;37(4):494-500. doi: 10.1002/mrm.1910370404. PMID: 9094070.

(4) Loegering IF, Denning SC, Johnson KM, Liu F, Lee KS, Thelen DG. Ultrashort echo time (UTE) imaging reveals a shift in bound water that is sensitive to sub-clinical tendinopathy in older adults. Skeletal Radiol. 2021 Jan;50(1):107-113. doi: 10.1007/s00256-020-03538-1. Epub 2020 Jul 8. PMID: 32642791; PMCID: PMC7677198.